# Inhibiting USP16 rescues stem cell aging and memory in an Alzheimer's model

Felicia Reinitz[1†], Elizabeth Y Chen[1†], Benedetta Nicolis di Robilant[1†], Bayarsaikhan Chuluun[2], Jane Antony[1], Robert C Jones[3], Neha Gubbi[1], Karen Lee[1], William Hai Dang Ho[1], Sai Saroja Kolluru[3], Dalong Qian[1], Maddalena Adorno[1], Katja Piltti[4], Aileen Anderson[4], Michelle Monje[1], H Craig Heller[2], Stephen R Quake[3], Michael F Clarke[1]*

[1]Institute of Stem Cell Biology and Regenerative Medicine, Stanford University School of Medicine, Stanford, United States; [2]Department of Biology, Stanford University, Stanford, United States; [3]Department of Bioengineering, Stanford University, Stanford, United States; [4]Sue and Bill Gross Stem Cell Research Center, University of California, Irvine, Irvine, United States

**Abstracts** Alzheimer's disease (AD) is a progressive neurodegenerative disease observed with aging that represents the most common form of dementia. To date, therapies targeting end-stage disease plaques, tangles, or inflammation have limited efficacy. Therefore, we set out to identify a potential earlier targetable phenotype. Utilizing a mouse model of AD and human fetal cells harboring mutant amyloid precursor protein, we show cell intrinsic neural precursor cell (NPC) dysfunction precedes widespread inflammation and amyloid plaque pathology, making it the earliest defect in the evolution of the disease. We demonstrate that reversing impaired NPC self-renewal *via* genetic reduction of USP16, a histone modifier and critical physiological antagonist of the Polycomb Repressor Complex 1, can prevent downstream cognitive defects and decrease astrogliosis in vivo. Reduction of USP16 led to decreased expression of senescence gene *Cdkn2a* and mitigated aberrant regulation of the Bone Morphogenetic Signaling (BMP) pathway, a previously unknown function of USP16. Thus, we reveal USP16 as a novel target in an AD model that can both ameliorate the NPC defect and rescue memory and learning through its regulation of both *Cdkn2a* and BMP signaling.

*For correspondence:
mfclarke@stanford.edu

†These authors contributed equally to this work

## Editor's evaluation

This work by Reinitz et al. provides nice evidence of a neural stem cell (NSC) defect that precedes and appears to be independent of amyloid pathology and neuroinflammation in an AD model. The authors show that targeting USP16, a BMI antagonist, can rescue some of these NSC deficits. Furthermore, scRNA-seq and GSEA points to the BMP pathway as a major player in regulating these phenotypes; in support of their hypothesis, the authors show that inhibition of BMP signaling using small molecules also rescues NSC defects. These are interesting and novel findings that will be of interest to the field.

## Introduction

Alzheimer's disease (AD) is the most common form of dementia, occurring in 10% of individuals over the age of 65 and affecting an estimated 5.5 million people in the United States (*Hebert et al., 2013*). Currently, there is no treatment to stop, prevent, or reverse AD, and recent advances with monoclonal antibody therapy targeting plaques, although controversial, might at best only slow progression

(*Huang and Mucke, 2012*; *Selkoe, 2019*; *Sevigny et al., 2016*). Historically, AD has been understood by its end-stage disease phenotype, characterized clinically by dementia and pathologically by amyloid senile plaques and neurofibrillary tangles (*Castellani et al., 2010*). These traditional AD pathologies are thought to begin with amyloid plaque deposition that is associated with inflammation, increased reactive oxygen species (ROS), and neurodegeneration during aging (*Akiyama et al., 2000*; *Glass et al., 2010*); however, thus far, treatments to decrease formation of plaques have shown only minimal or no improvement in disease progression or outcomes (*Knopman et al., 2021*; *Selkoe, 2019*).

Adult neurogenesis is thought to be compromised in AD, contributing to early dementia (*Alipour et al., 2019*). The decline of neural stem/precursor cell (NPC) function in the subventricular zone (SVZ) and the hippocampus has been established in both aging (*Leeman et al., 2018*) and various AD mouse models (*Haughey et al., 2002*; *López-Toledano and Shelanski, 2004*; *Mu and Gage, 2011*; *Rodríguez et al., 2009*; *Rodríguez and Verkhratsky, 2011*; *Sakamoto et al., 2014*; *Winner et al., 2011*). However, it is still unknown whether these defects are cell-intrinsic resulting from changes inside the cells or extrinsic, resulting from factors present in the niche, such as inflammation. Here, we report that the NPC defects seen in an AD mouse model harboring Swedish, Dutch, and Iowa mutations in the amyloid precursor protein (Tg-SwDI) is initially cell-intrinsic and predates inflammation and widespread plaque deposition, which play a role later in the disease. We chose the Tg-SwDI model with mutations confined to *APP* because mice develop early plaque deposition and cognitive deficits, and express physiologic levels of transgenic human AβPP at approximately 50% the level of endogenous mouse APP (*Davis et al., 2004*). In this model, mice begin to develop amyloid deposition in brain parenchyma as early as 3 months of age and throughout the forebrain by 12 months (*Davis et al., 2004*; *Miao et al., 2005*), as well as significant cerebral amyloid angiopathy, which is thought to be a more sensitive predictor of dementia than parenchymal and amyloid plaques (*Neuropathology Group. Medical Research Council Cognitive Function and Aging Study, 2001*; *Thal et al., 2003*).

In this study, we targeted USP16, an upstream regulator of *Cdkn2a*, to reverse the NPC defect seen in the Tg-SwDI model. USP16 counteracts self-renewal regulator BMI1 by deubiquitinating histone H2A on Lysine 119 at the *Cdkn2a* locus, resulting in increased expression of protein products P16 (*Ink4a*) and P14 (*Arf*) (*Adorno et al., 2013*). This results in increased senescence with a concomitant decrease in self-renewal. Here, we demonstrate that inhibiting USP16 is a potential novel strategy to rescue *Cdkn2a*-mediated pathologies in AD induced by both p16$^{Ink4a}$ and p19$^{Arf}$. As *Cdkn2a* might not be the only player in AD pathophysiology, we probed for additional pathways regulated by USP16 and identified enrichment of the BMP pathway early on in the Tg-SwDI mice. The BMP pathway has been known to play a role in NPC function. Specifically, BMPR2 is a type II receptor that heterodimerizes with BMPR1a or BMPR1b, and is responsible for transducing BMP signaling downstream to the SMAD proteins, which translocate to the nucleus and can turn on genes related to cell fate and differentiation (*Chang et al., 2018*). Levels of BMP2, 4, 6, and 7 expression have been found to increase in the hippocampus and SVZ with age (*Yousef et al., 2015*; *Apostolopoulou et al., 2017*). Furthermore, *Bmpr2* conditional ablation in *Ascl1* expressing neural stem cells (NSCs)/NPCs or treatment with BMP inhibitor Noggin results in activation of NSCs, increased cell proliferation, and a rescue of cognitive deficits to levels comparable to young mice (*Meyers et al., 2016*). For the first time, we show that targeting USP16 in a mouse model of AD rescues two aberrant aging pathways, *Cdkn2a* and BMP, which can restore self-renewal of NPCs, decrease astrogliosis, and retard cognitive decline (*Figure 1*).

## Results

### Neural precursor cell exhaustion is the earliest sign of disease in Tg-SwDI mice

Detecting disease early before fulminant pathogenesis may be crucial to develop effective diagnosis and treatment, particularly when it comes to irreversible degeneration. Therefore, we used a multimodal temporal approach, including immunofluorescence staining, in vitro neurosphere assays, Luminex assays, and behavioral studies to dissect changes at the molecular, cellular, and organismal levels in mice at varying ages. At 3 months of age, we found that proliferation of NPCs, marked by 5-ethynyl-2'-deoxyuridine (EdU) (*Chehrehasa et al., 2009*), SOX2, and GFAP, was increased threefold in the SVZ of Tg-SwDI mice (p=0.0153; *Figure 2A*). In many tissues including the blood, pancreas,

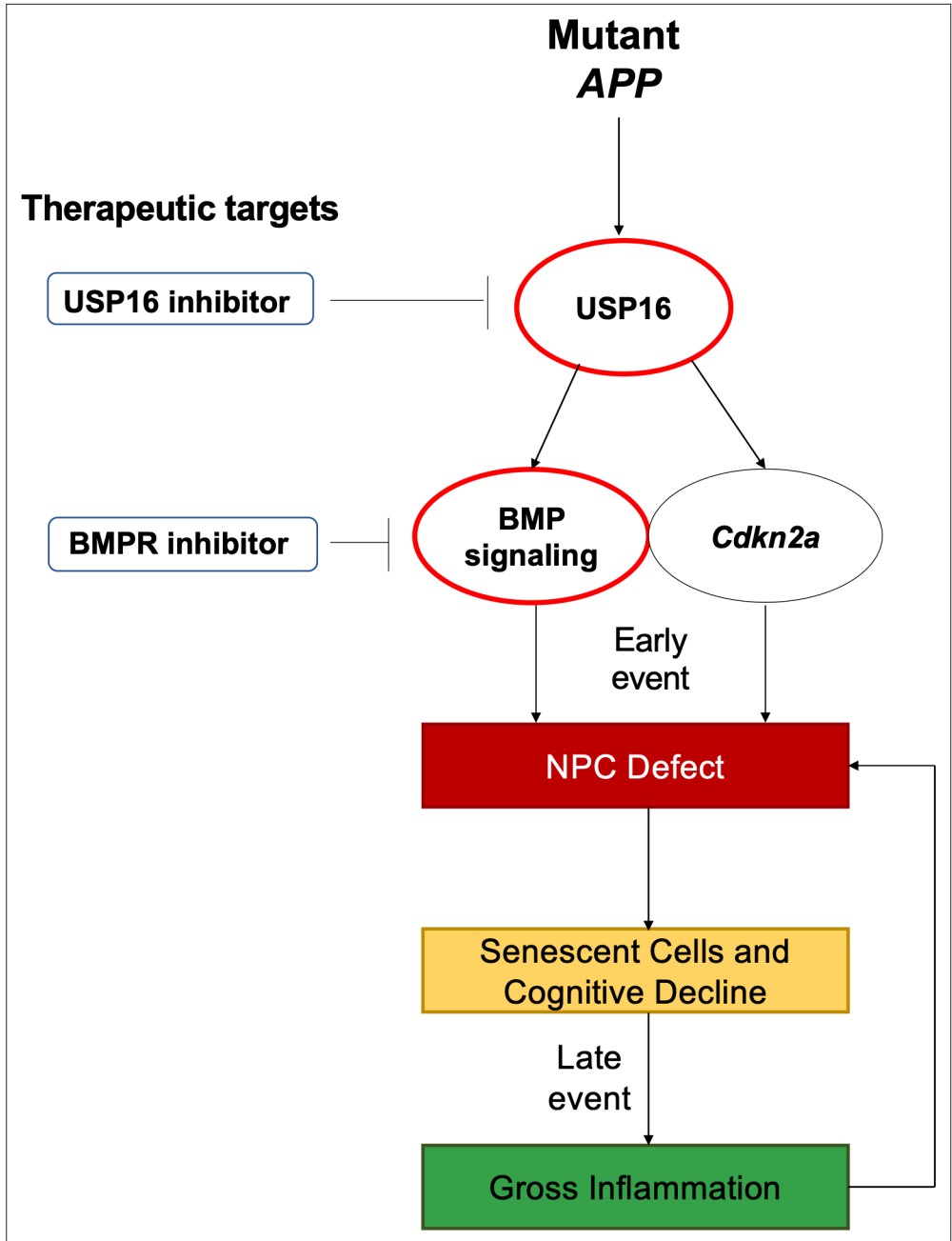

**Figure 1.** Schematic summarizing therapeutic approaches to mitigate the effects of mutant APP through targeting of *Cdkn2a*, BMI1, USP16 and BMP. Aberrant USP16/*Cdkn2a*/BMP signaling results in an early neural precursor cell (NPC) defect in Alzheimer's Disease. USP16 inhibitors and/or BMPR inhibitors can be combined with current therapeutics targeting beta amyloid plaques to rescue this earlier defect that predates senescence, cognitive decline, and resulting gross inflammation.

intestine, and mammary gland, hyperproliferation has been linked to a premature decline in stem cell function associated with aging (*Essers et al., 2009*; *Krishnamurthy et al., 2006*; *Scheeren et al., 2014*). Thus, we looked at stem cell function using extreme limiting dilution analysis (ELDA) of neurosphere-formation from single cells (*Hu and Smyth, 2009*; *Pastrana et al., 2011*). We discovered that 3 and 4-month-old Tg-SwDI mice had significantly less regenerative potential of the SVZ cells than that of healthy age-matched control mice (neurosphere-initiating cell (NIC) frequencies: 1 in 14.5 versus 1 in 7.5, respectively, p=0.00166, *Figure 2B* and *Figure 2—source data 1*). As *Bmi1* is required for self-renewal of stem cells in the peripheral and central nervous systems and is responsible

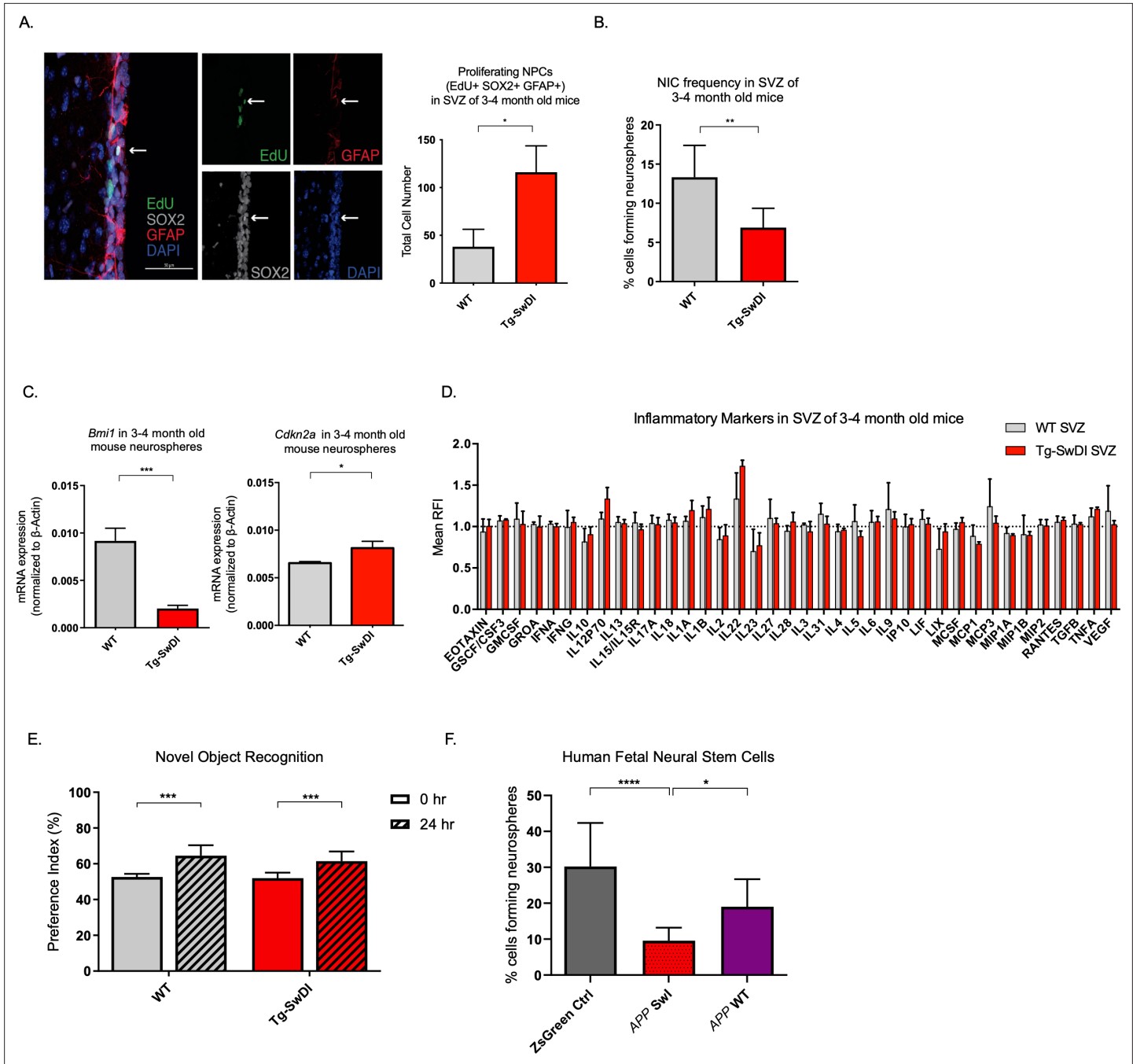

**Figure 2.** Defects in neurosphere initiating capacity (NIC) and hyperproliferation in Tg-SwDI mice predate cognitive deficits and widespread inflammation. (**A**) Representative 40× confocal images of the subventricular zone (SVZ) stains for EdU, GFAP, and SOX2 (left). Three to four-month-old mice underwent intraperitoneal injections every day for 6 days with EdU and the analysis was performed four weeks after. Count of proliferating neural precursor cells, as cells positive for EdU, GFAP, SOX2, and DAPI, is shown in the panel on the right (n = 3 mice). Data are presented as mean ± SEM. *p=0.0153 (**B**) Limiting dilution assays were performed using single cells derived from neurospheres from 3 to 4-month-old mice. The graph shows the percentage of neurosphere-initiating cells (NIC) ± upper and lower estimates converted to percentages from values calculated by extreme limiting dilution analysis (ELDA). **p=0.00166 *Figure 2—source data 2* summarizes the lower, upper, and estimates of 1/NIC for the different genotypes calculated by ELDA. (**C**) *Bmi1* and *Cdkn2a* expression levels measured by RT-qPCR in neurospheres derived from the SVZ of wild type (WT) or Tg-SwDI mice at third passage (mice aged 3 and 4 months). Data are presented as mean ± SD. ***p=0.0009 *p=0.0197 (**D**) Cytokine levels measured by Luminex array from the SVZ of young 3 and 4-month-old mice. No differences were observed at this age (n = 3 mice for each genotype). See also *Figure 2— figure supplement 1A*. Data are presented as mean ± standard deviation (SD). (**E**) Novel object recognition 24 hr testing in mice at 3 months of age showed no signs of cognitive impairment in the Tg-SwDI mice with a preference index comparable to that of WT indicating both genotypes had intact object discrimination (p=0.001 for WT and p=0.0099 for Tg-SwDI, n = 7–10 mice in each group). Data are presented as mean ± standard error of the

*Figure 2 continued on next page*

*Figure 2 continued*

mean (SEM). (**F**) ELDA graph of limiting dilution assay comparing human fetal neurospheres infected with pHIV-Zsgreen, *APP* SwI, or *APP* WT. **Figure 2—source data 2** lists the estimated stem cell frequencies and ranges for each group, calculated using the ELDA software (n = 3 separate transductions and limiting dilution experiments) ****p=3.7e$^{-6}$, *p=0.00507.

The online version of this article includes the following source data and figure supplement(s) for figure 2:

**Source data 1.** NIC frequencies in 3-4 month old mice.

**Source data 2.** NIC frequencies in fetal cells.

**Figure supplement 1.** No significantly upregulated inflammatory markers in young Tg-SwDI mice with at least a 2-fold increase in transgenic mutant *APP* Swedish Dutch Iowa (SDI) expression.

for repressing p16$^{Ink4a}$ expression, we queried if decreased *Bmi1* and/or increased *Cdkn2a* expression was concurrent with the observed decrease in regenerative potential of NPCs from Tg-SwDI mice (*Molofsky et al., 2005*). Indeed, we measured a significant decrease in *Bmi1* expression and a significant increase in its downstream target *Cdkn2a* expression in neurospheres from Tg-SwDI mice compared to wild type (WT) (*Figure 2C*). To study whether these changes in proliferation and self-renewal capacity of NPCs occur before the well-established prominent AD phenotype of inflammation, we employed a Luminex screen to assess the presence of an array of cytokines and other inflammatory markers. We looked at the SVZ, hippocampal dentate gyrus (DG), and cortex in 3 and 4-month-old mice, but found no significant differences in inflammatory markers between Tg-SwDI and WT mice in any of these regions (*Figure 2D* and *Figure 2—figure supplement 1A*). To explore further, we measured mRNA levels of *Ptgs2* (COX2), *Tnf, Il6,* and *Il1b* utilizing quantitative polymerase chain reaction (qPCR), but found no significant differences between WT and Tg-SwDI in either the SVZ, DG, or cortex for *Ptgs2*, whereas the remaining markers were undetectable using quantitative RT-PCR (*Figure 2—figure supplement 1B*).

One of the hypothesized reasons for lack of efficacy observed in AD clinical trials is the late initiation of treatment. In humans, abnormal deposits of amyloid β and tau tangles as well as damage of the brain is believed to start a decade or more before cognitive decline (*Ower et al., 2018*). We, therefore, wanted to see if the decrease in neurosphere-initiating capacity of Tg-SwDI mice also precedes their memory impairment and progressively diminished cognitive function. Although Tg-SwDI mice are known to exhibit these features, there was no evidence of cognitive impairment in 3 and 4-month-old mice when subjected to novel object recognition (NOR) training and subsequent testing after 24 hr (*Ennaceur and Delacour, 1988*; *Figure 2E*).

Finally, given the prominent early aberrant self-renewal phenotype in the 3 and 4-month-old Tg-SwDI mice, we investigated whether or not expression of mutant *APP* in human NPCs might also cause a self-renewal defect. To this end, we infected human fetal neurospheres with a lentiviral construct for either pHIV-Zsgreen alone, pHIV-Zsgreen with wild type *APP* (*APP* WT), or pHIV-Zsgreen with Swedish and Indiana *APP* mutations (*APP* SwI). This model allowed us to study the potentially cell intrinsic effect of *APP* mutations on NPC regenerative potential in primary human cells. Although our Tg-SwDI mice employ Swedish, Dutch, and Iowa mutations, we chose a human in vitro model with Swedish and Indiana mutations as the Indiana mutation allows for an increase in the Aβ$_{42}$/Aβ$_{40}$ ratio rather than the increased amyloidosis of the vasculature incurred by Dutch and Iowa mutations which would have been difficult to observe in vitro. Our human NPCs expressed at least a 2-fold change increase in mutant *APP* compared to the baseline endogenous *APP* levels in the Zsgreen control (*Figure 2—figure supplement 1C*). Employing the same limiting dilution assay as before, we found diminished NIC frequency of mutant *APP*-infected human neurospheres compared to cells infected with the empty vector or with WT *APP* (1 in 10.44 versus 1 in 3.31 and 1 in 10.44 versus 1 in 5.26, respectively, p=3.7e$^{-06}$ and p=0.00507; *Figure 2F* and *Figure 2—source data 1*). This result suggests that the self-renewal defect is cell-intrinsic and can be observed in two different AD models with different mutations.

## Modest aging in Tg-SwDI accelerates NPC exhaustion prior to detectable inflammation

To explore progression of the disease with aging, we next looked at what phenotypic changes occurred in older Tg-SwDI mice, including proliferation, self-renewal, inflammation, and astrogliosis.

The NPC hyperproliferation in SVZ observed in 3 and 4-month-old Tg-SwDI mice was not observed in 1-year-old mice, demonstrated by the number of EdU+SOX2+GFAP+ cells (*Figure 3A*). Still, the defect in self-renewal that was observed in the 3 and 4-month-old Tg-SwDI mice was exacerbated in the 1-year-old Tg-SwDI mice (p=0.00625; *Figure 3B* and *Figure 3—source data 1*).

We hypothesized that inflammation might explain the NPC defect but may not have been easily detected at 3 months of age. However, even at 1 year old, we did not detect any overall significant differences in inflammatory cytokines in the SVZ, DG, or cortex between the WT and Tg-SwDI mice (*Figure 3C*, *Figure 3—figure supplement 1A, B*). Reactive astrogliosis, the abnormal increase and activation of astrocytes seen in AD patients and mouse models, is also a sign of inflammation that can drive degeneration of neurons and has been linked to both AD disease pathogenesis (*Osborn et al., 2016*) and to the BMI1/*Cdkn2a* pathway. Specifically, Zencak and colleagues showed increased astrogliosis in *Bmi1*[-/-] mice (*Zencak et al., 2005*). With the aim of evaluating astrogliosis, we performed a qPCR for A1 astrocytic markers in the cortex of 1-year-old mice, but did not observe any significant increases in mRNA expression compared to WT (*Figure 3—figure supplement 2*).

However, when we looked at differentiation of neurosphere cultures from the SVZ of Tg-SwDI mice and WT controls and analyzed the number of GFAP-expressing cells following differentiation, we found that cells originating from Tg-SwDI mice formed more GFAP+ cells than those from WT controls. This suggested that there was a general lineage increase in astrocytes derived from Tg-SwDI NPCs compared to WT NPCs (*Figure 3—figure supplement 3*).

Because inflammation and reactive astrogliosis are linked to AD, we wished to see when these events occur in our models. When we looked at 2-year-old mice by microarray analyses of SVZ, DG, and cortex, we saw an increase in *Cd44*, *Vim*, *Serping1,* and other markers related to pan- and A1-specific astrogliosis and a concomitant inflammatory signature of Tg-SwDI mice compared to WT mice (*Figure 3—figure supplement 4A, B*, respectively, *Reinitz et al., 2022*). We next looked for additional evidence of astrogliosis and observed a significant increase in GFAP-expressing cells in the cerebral cortex of 2-year-old Tg-SwDI mice compared to WT that was not seen in younger Tg-SwDI mice (p<0.0001; *Figure 3D* and *Figure 3—figure supplement 5*). This suggests that reactive astrogliosis is exacerbated by and strongly correlated with aging and later disease progression.

In line with our findings thus far, expression of the well-studied gene, *Cdkn2a*, known for its increased expression with aging and critical function of inhibiting stem cell self-renewal during development and throughout the lifespan, was increased with aging and even more so in the Tg-SwDI cortex (*Figure 3E*). (*Molofsky et al., 2003*; *Park et al., 2003*; *Sun et al., 2004*). Taken together with our data showing an increase in EdU+SOX2+GFAP+ cells in Tg-SwDI mice SVZ at an early age, our extreme limiting dilution assay in older mice, and expression changes of *Cdkn2a* and *Bmi1*, we infer a premature reduction of self-renewal capacity of NPCs, but not necessarily a decrease in the total number of neural progenitor cells in our Tg-SwDI model. These results suggest that a neural stem cell defect and early cognitive decline predate detectable inflammation and reactive astrogliosis in Tg-SwDI mice.

## Self-renewal defects are rescued by *Usp16* and *Cdkn2a* modulation

Neural precursor cells function through a number of genetic and epigenetic components, and one of the well-described master regulators is *Cdkn2a,* a gene tightly regulated by BMI1 (*Bruggeman et al., 2005*). When we crossed the Tg-SwDI mouse with a *Cdkn2a* knockout mouse (Tg-SwDI/*Cdkn2a*[-/-]) and performed limiting dilution assays in SVZ cells from 3-month-old mice, there was a complete restoration of the NIC frequency in the Tg-SwDI/*Cdkn2a*[-/-] cells compared to age-matched Tg-SwDI cells (p=7.7e[−05]; *Figure 4A*, *Figure 4—source data 1*, and *Figure 4—figure supplement 1A*, *Figure 4—figure supplement 1—source data 1*). This NIC rescue was also observed in hippocampal cells cultured from microdissection of the DG (p=2.09e[−9], *Figure 4B* and *Figure 4—source data 2*). These results demonstrate that impairment of NPC regeneration, as measured by NIC frequencies, is a function of aging that is accelerated by *APP* mutations and is mitigated through loss of *Cdkn2a*, a known regulator of NPC self-renewal (*Molofsky et al., 2003*).

Unfortunately, mutations or loss of function in the *Cdkn2a* gene eventually leads to tumor formation, making it not feasible to perform limiting dilution experiments in 1-year-old *Cdk2na* knockout mice and also making it less than ideal to target therapeutically (*Hussussian et al., 1994*). Upstream of *Cdkn2a* is USP16, an antagonist of BMI1 and a de-repressor of *Cdkn2a* that acts through the

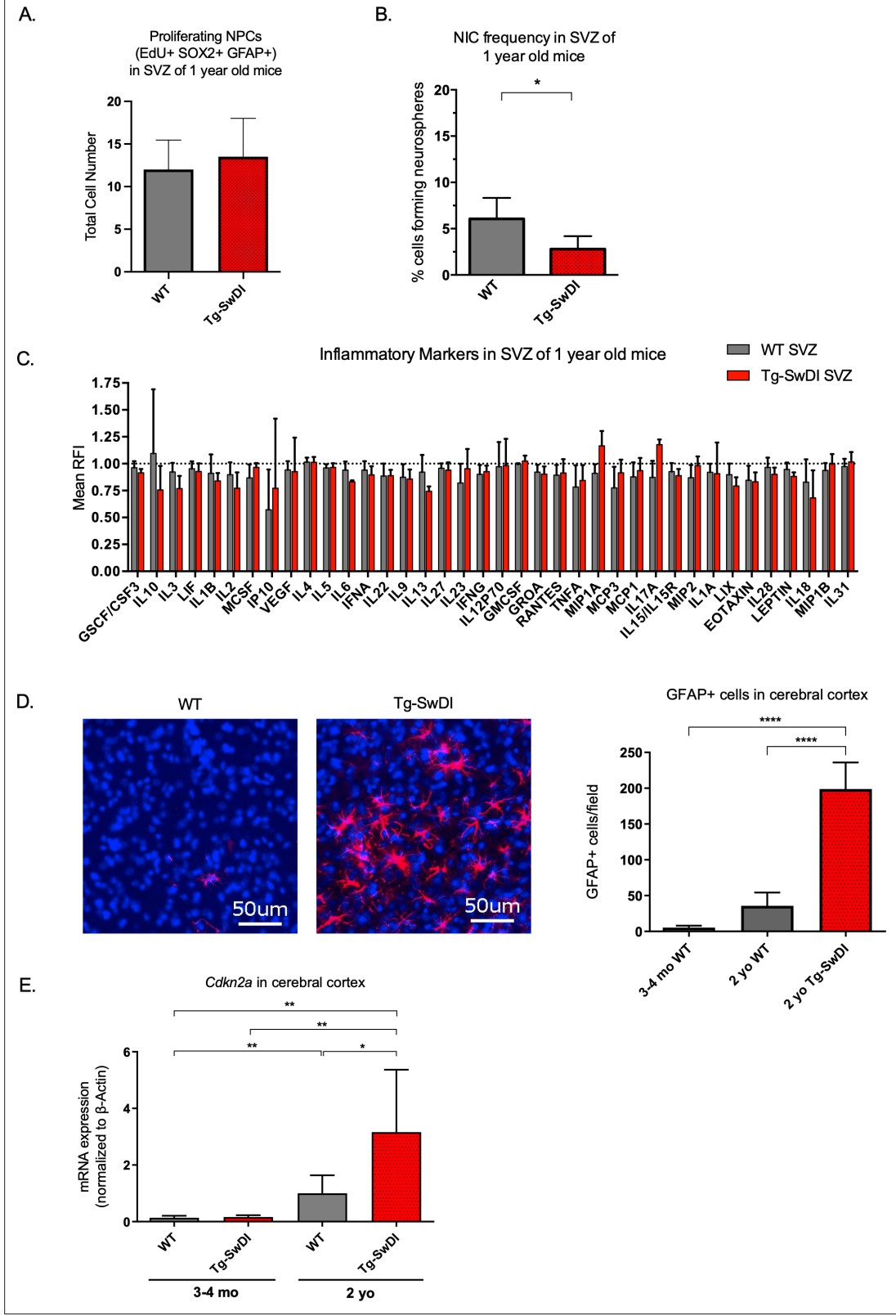

**Figure 3.** Accelerated aging phenotype seen in Tg-SwDI mice with exacerbated self-renewal and astrogliosis. (A) One-year-old mice underwent intraperitoneal injections every day for 6 days with EdU and the analysis was performed four weeks afterward to capture all true activated daughter stem cells that would maintain the niche without further differentiation or migration. Data are presented as counts of proliferating neural precursor cell cells positive for EdU, GFAP, SOX2, and DAPI mean ± SEM (n = 3 mice). (B) Limiting dilution assays were performed using single cells derived from

*Figure 3 continued on next page*

*Figure 3 continued*

neurospheres from 1-year-old mice. The bar graph shows the percentage of neurosphere-initiating cells calculated by extreme limiting dilution analysis (ELDA). *p=0.00625. *Figure 3—source data 1* summarizes the lower, upper, and estimates of 1/NIC for the different genotypes calculated by ELDA. (C) Cytokine levels measured by Luminex array from the subventricular zone of 1-year-old mice. No differences were observed at this age. (n = 3 mice each genotype). See also *Figure 3—figure supplement 1*. Data are presented as mean ± SD. (D) Anterior sections were obtained from 2-year-old mice, stained and counted for GFAP+ cells in the cortex. Four different images per section and three sections per mouse were counted (n = 4 mice each group). A one-way ANOVA showed significant differences between the groups (****p<0.0001). Data are presented as mean ± SEM. (E) mRNA levels of *Cdkn2a* in the cerebral cortex of 3 and 4-month-old and 2-year-old mice were measured by RT-qPCR. Ct values were normalized to *Actb*. (WT = wild type littermate; 3 and 4-month-old: WT = 7, Tg-SwD n = 7, 2-year-old: WT n = 6, Tg-SwDI n = 6). A one-way ANOVA showed significant differences between the groups (p=0.0044 between 3 and 4-month-old and 2-year-old WT, p=0.0040 between 3 and 4-month-old and 2-year-old Tg-SwDI, p = 0.0438 between 2-year-old WT and Tg-SwDI, p=0.00553 between 3 and 4-month-old WT and 2-year-old Tg-SwDI). Data are presented as mean ± SD.

The online version of this article includes the following source data and figure supplement(s) for figure 3:

**Source data 1.** NIC Frequncies in 1 yo mice.

**Figure supplement 1.** No significantly upregulated inflammatory markers in 1-year-old Tg-SwDI mice.

**Figure supplement 2.** No significant increase in levels of A1 astrocyte markers at 1 year of age in Tg-SwDI mice.

**Figure supplement 3.** There is a general lineage increase in astrocytes derived from Tg-SwDI neural precursor cells.

**Figure supplement 4.** Upregulation of astrogliosis and a concomitant inflammatory signature is present in Tg-SwDI mice compared to wild type (WT) mice.

**Figure supplement 5.** There is no upregulation of astrogliosis in 3 and 4-month-old Tg-SwDI mice over wild type (WT) mice.

enzymatic removal of ubiquitin from histone H2A (*Figure 4C*; *Adorno et al., 2013*; *Joo et al., 2007*). We predicted that downregulation of *Usp16* would increase BMI1 function to counteract the effects of mutant APP similar to what we observed with knockout of *Cdkn2a*. This is supported by previous data that showed overexpression of *USP16* in human-derived neurospheres led to a marked decrease in the formation of secondary neurospheres (*Adorno et al., 2013*), and overexpression of *Bmi1* led to increased self-renewal and maintenance of multipotency (*Fasano et al., 2009*). To test this, we crossed Tg-SwDI mice with *Usp16*⁺ᐟ⁻ mice to generate Tg-SwDI/*Usp16*⁺ᐟ⁻ mice, which do not show tumor formation. We found that Tg-SwDI mice express greater than twofold more cortical *Cdkn2a* than both WT and Tg-SwDI/*Usp16*⁺ᐟ⁻ mice, for which expression levels were very similar (p=0.0365 and p=0.0318, respectively, *Figure 4D*). Limiting dilution experiments of cells isolated from the SVZ and DG of the hippocampus showed that Tg-SwDI/*Usp16*⁺ᐟ⁻ mice had significantly greater NIC frequencies, partially rescuing the self-renewal defect seen with mutant APP (p=0.0492 and p=0.00233, respectively; *Figure 4E*, *Figure 4—source data 3* and *Figure 4F*, *Figure 4—source data 4*). Furthermore, two-year-old Tg-SwDI mice show reduced *Bmi1* expression, which was rescued by *Usp16* haploinsufficiency, in both the SVZ and cortex (*Figure 4—figure supplement 1B*, *Reinitz et al., 2022*). Similar to the NIC rescue in the Tg-SwDI/*Cdkn2a*⁻ᐟ⁻ mice, these data provide further evidence of cell-intrinsic impaired self-renewal in the Tg-SwDI model of familial AD, and that reversal of this impairment is possible through targeting *Cdkn2a* upstream regulator, USP16.

## RNA-seq reveals enriched BMP signaling in Tg-SwDI mice that is rescued by *Usp16* haploinsufficiency

Previous studies using RNA-sequencing techniques have demonstrated significant genomic age-related cell intrinsic changes in self-renewing cells originating from the SVZ (*Apostolopoulou et al., 2017*). To delineate potential self-renewal pathways that might contribute to the defect and rescue of Tg-SwDI NPCs and Tg-SwDI/*Usp16*⁺ᐟ⁻ NPCs, respectively, we performed single-cell RNA-seq on lineage depleted primary FACS-sorted CD31⁻CD45⁻TER119⁻CD24⁻ SVZ cells from Tg-SwDI, WT, and Tg-SwDI/*Usp16*⁺ᐟ⁻ mice at 3 and 4 months and 1 year of age (*Figure 5A*; *Mootha et al., 2003*; *Subramanian et al., 2005*). As CD31/CD45/TER119 denote the hematopoietic cell fraction and CD24 marks differentiated cells, we used these markers to enrich for NPCs obtained from the SVZ. A cell-type analysis using a TSNE plot surveying the top differentially expressed genes of each cluster (*Tabula Muris Consortium et al., 2018*; *Zhang et al., 2014*) did not find any new cell populations specific to the Tg-SwDI genotype (*Figure 5—figure supplement 1A, B*, *Chen et al., 2022*). This suggested to us that changes in phenotype were the result of transcriptional differences in cells rather than the addition or subtraction of an existing cell type. Like our human model, cells from the Tg-SwDI mouse

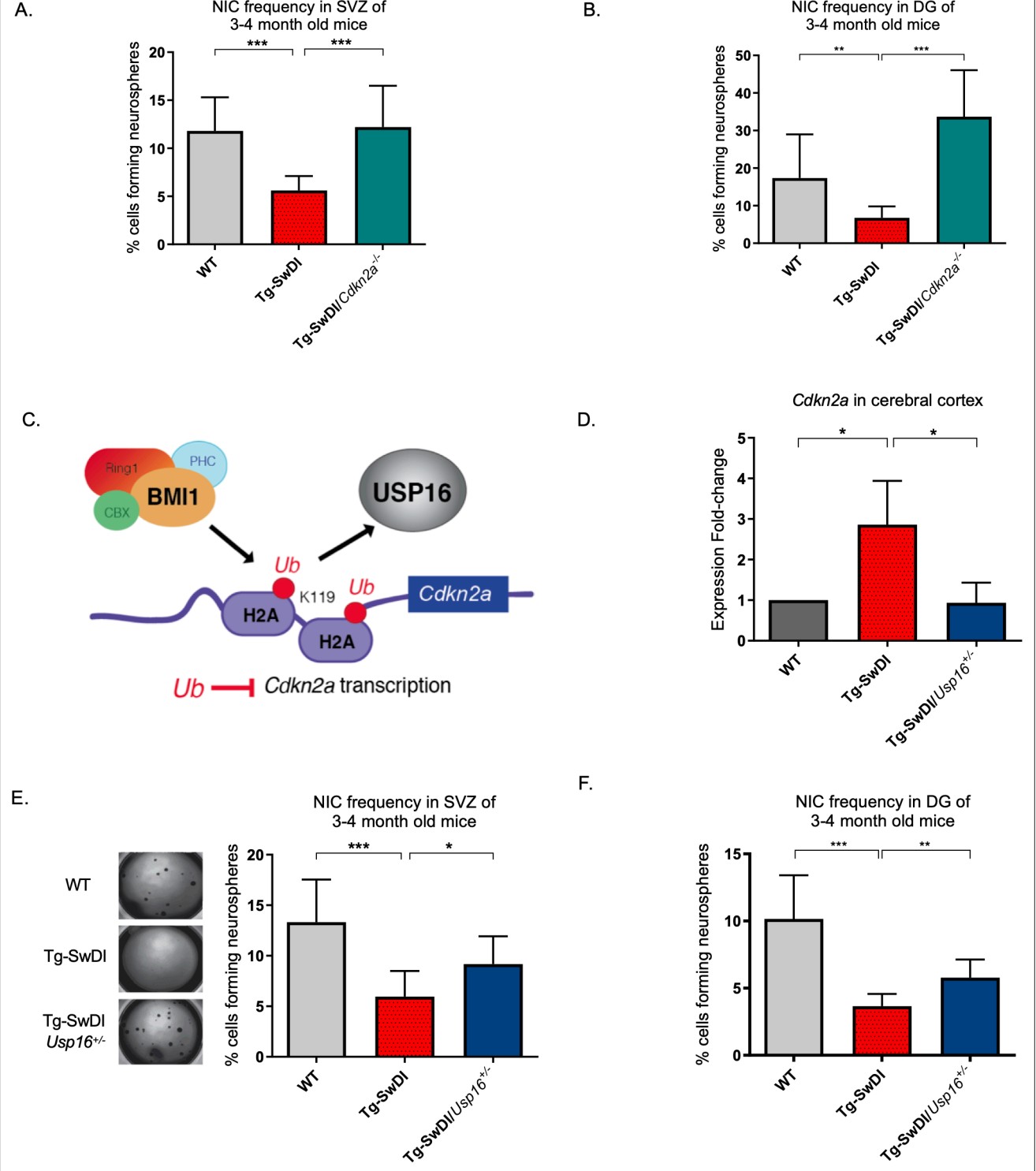

**Figure 4.** *Usp16* haploinsufficiency normalizes *Cdkn2a* expression and restores self-renewal in Tg-SwDI NPCs. (**A**) The bar graph shows the NIC frequencies in subventricular zone (p=5.5e$^{-5}$ between wild type (WT) and Tg-SwDI and p=7.7e$^{-5}$ between Tg-SwDI/*Cdkn2a*$^{-/-}$ and Tg-SwDI) and in the dentate gyrus (**B**) (p=0.00476 between wild type (WT) and Tg-SwDI and p=2.09e$^{-9}$ between Tg-SwDI/*Cdkn2a*$^{-/-}$ and Tg-SwDI) as percentages of total cells with error bars indicating the upper and lower values. Mice were 3 months old when sacrificed; experiment done after third passage of neurospheres. , *Figure 3—source data 1*, *Figure 4—source data 1* summarize the lower, upper, and estimates of 1/NIC for the different genotypes calculated by extreme limiting dilution analysis (ELDA). (**C**) Schematic summarizing the role of BMI1 in ubiquitinating histone H2A at different sites in the genome, including the *Cdkn2a* locus and the role of USP16 as its natural antagonist, suggesting that USP16 inhibition could influence neurosphere initiating

*Figure 4 continued on next page*

*Figure 4 continued*

capacity. (**D**) RT-qPCR of *Cdkn2a* in the cerebral cortex of 2-year-old Tg-SwDI mice shows mRNA levels were rescued by *Usp16* haploinsufficiency (n = 3). Ct-values were normalized to *Actb*. A one-way ANOVA showed significant differences between the groups (p=0.0365 between WT and Tg-SwDI and p=0.0318 between Tg-SwDI and Tg-SwDI/*Usp16*⁺ᐟ⁻). Data are presented as mean ± SD. (**E**) Left panel shows 1× representative photographs of neurospheres grown in 96-well dish after two weeks of culture. The bar graph shows the NIC frequencies in subventricular zone as percentages of total cells comparing WT, Tg-SwDI, and Tg-SwDI/*Usp16*⁺ᐟ⁻ mice. Mice were 3 months old. (n = 3 mice per genotype, p=0.000402 between WT and Tg-SwDI and p=0.0492 between Tg-SwDI/*Usp16*⁺ᐟ⁻ and Tg-SwDI) (**F**) The bar graph shows the NIC frequencies in dentate gyrus as percentages of total cells comparing WT, Tg-SwDI, and Tg-SwDI/*Usp16*⁺ᐟ⁻ mice. (n = 3 mice per genotype, p=9.9e⁻⁹ between WT and Tg-SwDI and p=0.00233 between Tg-SwDI/*Usp16*⁺ᐟ⁻ and Tg-SwDI) *Figure 4—source data 3* and *Figure 4—source data 4* summarize the lower, upper, and estimates of 1/NIC for the different genotypes calculated by ELDA.

The online version of this article includes the following source data and figure supplement(s) for figure 4:

**Source data 1.** NIC Frequencies *Cdkn2a* in SVZ.

**Source data 2.** NIC Frequencies *Cdkn2a* in DG.

**Source data 3.** NIC Frequencies *Usp16* in SVZ.

**Source data 4.** NIC Frequencies *Usp16* in DG.

**Figure supplement 1.** *Cdkn2a* loss increases NIC capacity while *Usp16* haploinsufficiency rescues *Bmi1* expression in Tg-SwDI mice.

**Figure supplement 1—source data 1.** NIC Frequencies *Cdkn2a* control.

model expressed approximately a 1.5-fold increase in both normalized mutant *APP* expression and endogenous *App* expression compared to WT cells (*Figure 5—figure supplement 2A*, *Figure 5—figure supplement 2—source data 1*, *Chen et al., 2022*). In addition, similar to the Luminex screen, we did not observe any significant transcriptional upregulation of an inflammatory signature in either 3 and 4-month-old or 1-year-old mice between Tg-SwDI and WT mice (*Figure 5—figure supplement 2B, C*, *Figure 5—figure supplement 2—source data 2* and *Figure 5—figure supplement 2—source data 3*, *Chen et al., 2022*). It is important to note that the RNA-seq data presented here were conducted on cells enriched in the SVZ, which does not exclude the possibility of inflammation in other areas of the brain. Furthermore, when we looked at a panel of proliferation-related genes (*Figure 5—figure supplement 3A, B*, *Figure 5—figure supplement 3—source data 1* and *Figure 5—figure supplement 3—source data 2*, *Chen et al., 2022*), there were only a few genes significantly different between WT and Tg-SwDI cells at 3 and 4 months or 1 year of age, suggesting that the hyperproliferation phenotype we observed in 3 and 4-month-old mice might involve upregulation of only a few genes such as *Anapc2*, *Cdk4*, and *Pcna* and downregulation of cell cycle inhibitors *Cdkn1a* and *Cdkn1b*. This analysis was limited by the fact that the same sorted cells in our scheme may not be equivalent to the cells stained for EdU, SOX2, and GFAP in *Figure 2A*. Other significantly upregulated and downregulated genes are shown labeled on the volcano plots in *Figure 5—figure supplement 4A, B*.

Oftentimes, when we are looking at individual differentially expressed genes, it can be difficult to select which genes to pursue in a therapeutic context. We therefore performed gene set enrichment analysis (GSEA) in order to highlight groups of genes that are significantly enriched and related in the same pathway and can be more easily targeted. Using the GSEA Hallmark gene sets, we found only three gene sets that were enriched in Tg-SwDI mice over WT mice and were also rescued in the Tg-SwDI/*Usp16*⁺ᐟ⁻ mice at both ages: TGF-ß pathway, oxidative phosphorylation, and *Myc* Targets (*Figure 5—source data 1*). The TGF-ß pathway consistently had the highest normalized enrichment score in pairwise comparisons between Tg-SwDI versus WT and Tg-SwDI versus Tg-SwDI/*Usp16*⁺ᐟ⁻ of the three rescued pathways (*Figure 5—source data 2*). In looking specifically at the leading-edge genes contributing to the enrichment plots of the TGF-ß pathway, we found upregulation of BMP receptors and *Id* genes, which are known to be involved in BMP signaling, a sub-pathway of TGF-ß (*Figure 5B*). Heatmaps of average normalized single-cell gene expression showed BMP receptors as the most highly expressed TGF-ß receptors, with genes such as *Bmpr2*, *Bmpr1a*, *Id2*, and *Id3* upregulated in Tg-SwDI mice and rescued in Tg-SwDI/*Usp16*⁺ᐟ⁻ mice (*Figure 5C*). Furthermore, the BMP response *Id* genes showed stronger localization to the SLC1A3⁺ NPC clusters of both 3 and 4-month-old and 1-year-old mice than genes of the oxidative phosphorylation and *Myc* target pathways (*Figure 5—figure supplements 5 and 6*). These data suggest that USP16 may regulate NPC function in part through the BMP pathway.

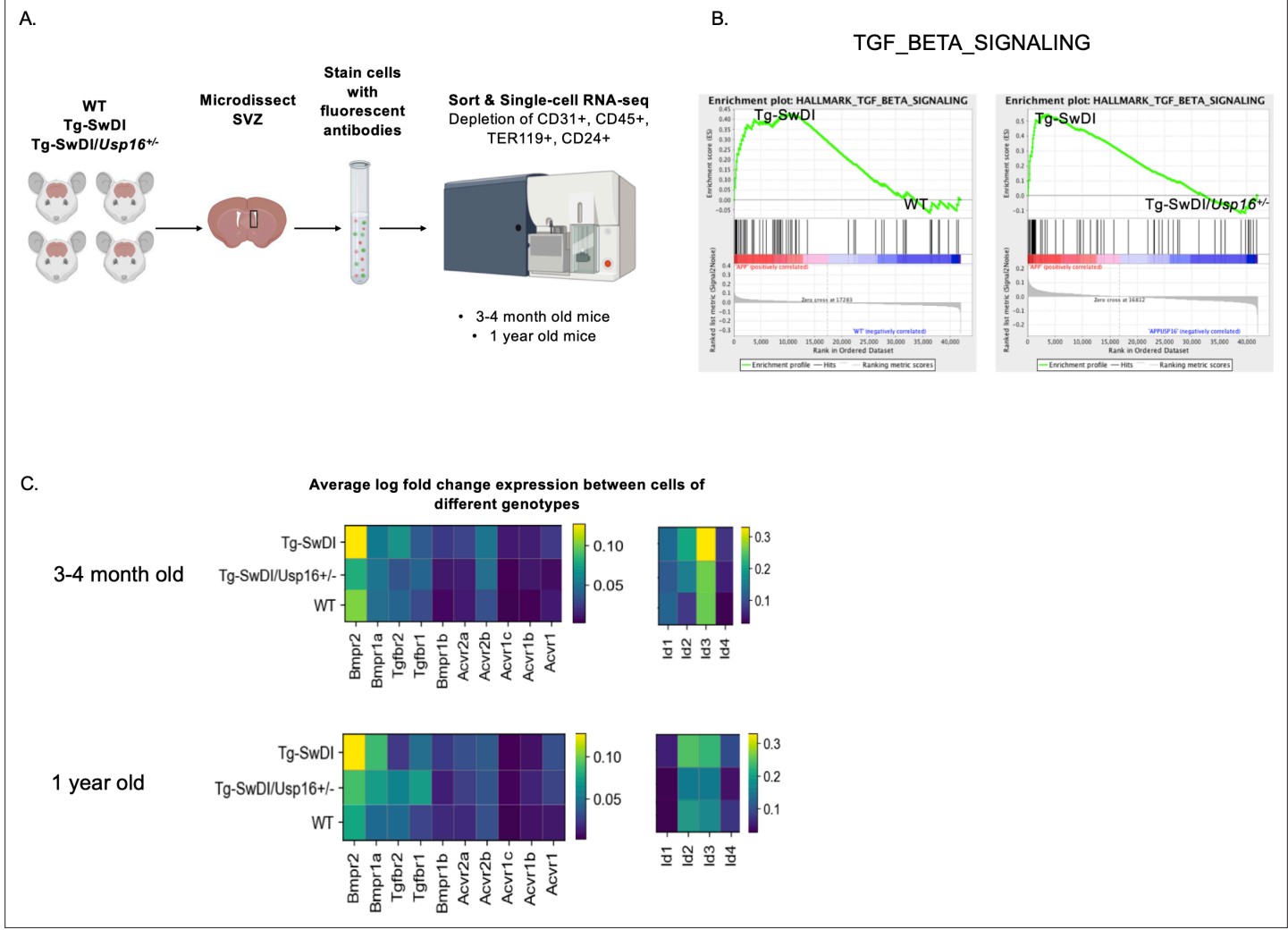

**Figure 5.** BMP signaling is enriched in Tg-SwDI mice and decreases with *Usp16* haploinsufficiency. (**A**) Lineage⁻CD24⁻ NPCs were FACS-sorted from the subventricular zone of 4 mice each of the different genotypes and processed for single-cell RNA-sequencing. *Figure 5—source data 1*: GSEA analysis from single-cell RNA-seq data shows pathways enriched in Tg-SwDI mice compared to wild type (WT) and rescued in Tg-SwDI/*Usp16⁺/⁻* mice, ordered top to bottom from smallest FDR q-val (most significant) to largest FDR q-val (least significant). (n = 4 for each genotype at each time point; FDR < 25%). Pathways in common to both age groups are bolded. *Figure 5—source data 2*: Normalized enrichment scores of significantly enriched pathways in Tg-SwDI mice compared to WT or Tg-SwDI/*Usp16⁺/⁻* mice at different time points. TGFß pathway, Oxidative phosphorylation, and MYC Targets V2 were selected as they were rescued in both 3 and 4 months and 1-year-old mice by *Usp16* haploinsufficiency. Highest normalized enrichment scores of each comparison are bolded. (**B**) Enrichment plots show TGF-ß signaling pathway as enriched in Tg-SwDI mice and rescued by *Usp16* haploinsufficiency. Normalized enrichment score (NES) for left panel is 1.77 with FDR-q value = 0.008; NES for right panel is 2.30 with FDR q-value <0.001. (**C**) Heatmaps showing averaged log-normalized single-cell gene expression of elements of the TGF-ß pathway; elements of the BMP pathway, a sub-pathway of the TGF-ß pathway, are specifically enriched in Tg-SwDI mice.

The online version of this article includes the following source data and figure supplement(s) for figure 5:

**Source data 1.** Pathways enriched in Tg-SwDI and rescued in Tg-SwDI/*Usp16⁺/⁻* mice.

**Source data 2.** Normalized Enrichment Scores of Significantly Enriched Pathways.

**Figure supplement 1.** Comprehensive transcriptomic analysis of Lin-CD24- NPCs from subventricular zone (SVZ) of 3 and 4-month-old and 1-year-old mice show no new cellular subpopulations in Tg-SwDI or Tg-SwDI/*Usp16⁺/⁻* mice.

**Figure supplement 2.** Few changes in inflammatory markers in Tg-SwDI or wild type (WT) mice at 1-year of age with an approximately 1.5-fold increase in normalized mutant *APP* expression.

**Figure supplement 2—source data 1.** *App* gene expression in WT and Tg-SwDI mice.

**Figure supplement 2—source data 2.** Inflammatory genes in 3-4 month old mice.

**Figure supplement 2—source data 3.** Inflammatory genes in 1 year old mice.

*Figure 5 continued on next page*

*Figure 5 continued*

**Figure supplement 3.** Only a few hyperproliferation genes are significantly different between wild type (WT) and Tg-SwDI cells at 3 and 4 months or 1 year of age.

**Figure supplement 3—source data 1.** Cell cycle related genes in 3-4 months old mice.

**Figure supplement 3—source data 2.** Cell cycle related genes in 1 yo mice.

**Figure supplement 4.** Differentially expressed genes in Tg-SwDI cells compared to wild type (WT) cells.

**Figure supplement 5.** TSNE plots of 3 and 4-month-old mice showing localization of various genes in pathways highlighted in *Figure 5—source data 2*.

**Figure supplement 6.** TSNE plots of 1-year-old mice showing localization of various genes in pathways highlighted in *Figure 5—source data 2*.

## BMPR inhibition rescues stem cell defects and abolishes increased phospho-SMAD 1/5/8

To confirm the functional significance of the BMP pathway in *APP*-mediated self-renewal defects, we measured the effects of modulating BMP pathway activity in vitro in human fetal NPCs expressing *APP* with Swedish and Indiana mutations (*APP* SwI). First, we measured levels of phosphorylated-SMAD (pSMAD) 1, 5, and 8, known downstream regulators of BMP activity, and found they were significantly increased in the mutant neurospheres compared to control (p=0.0001, *Figure 6A*). Treatment of the neurospheres with the BMP receptor inhibitor LDN-193189, a specific inhibitor of BMP-mediated SMAD1, SMAD5, and SMAD8 activation, substantially decreased pSMAD 1/5/8 in *APP* SwI NPCs

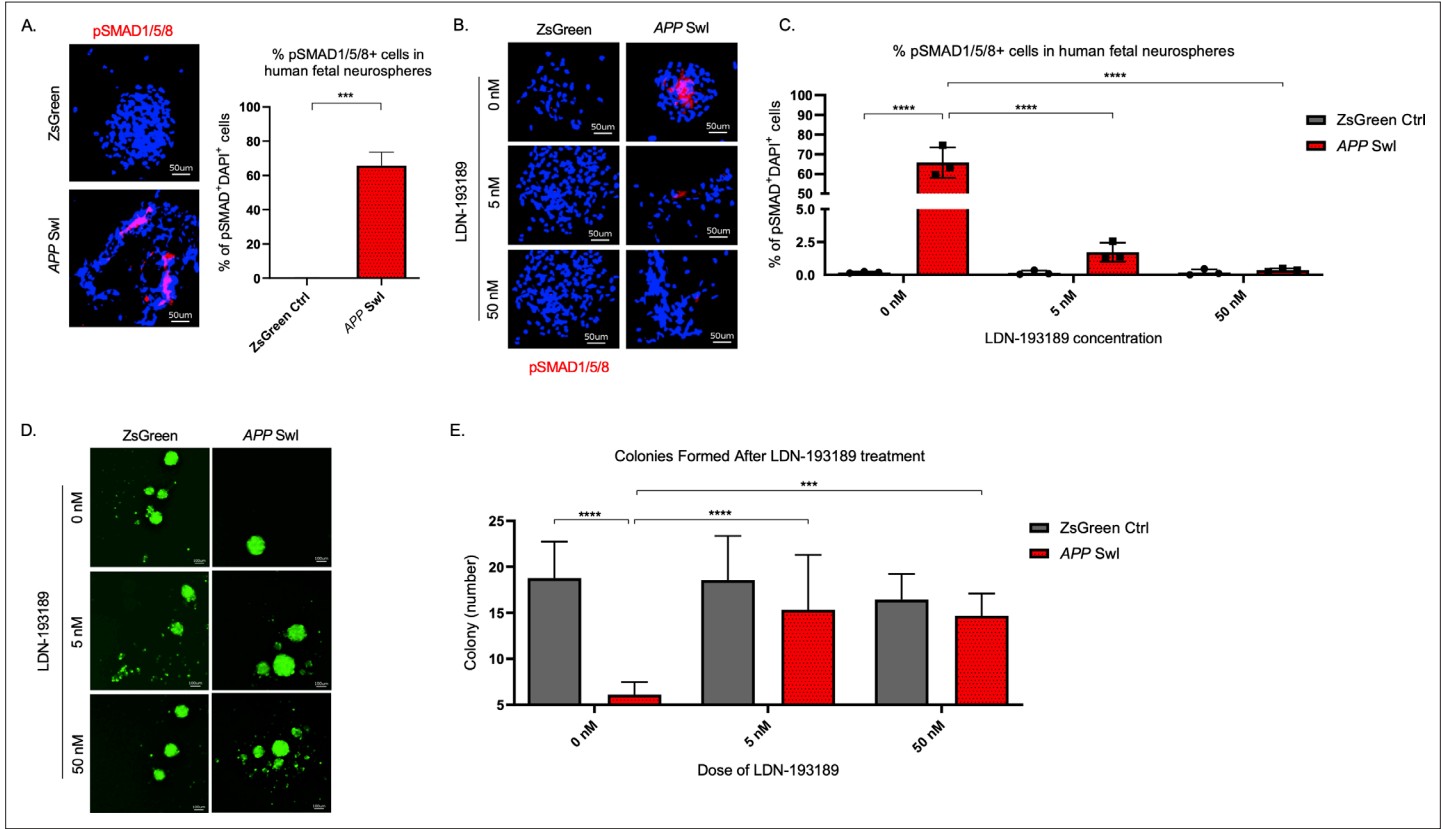

**Figure 6.** BMPR inhibition rescues mutant *APP* mediated self-renewal defects in human neurospheres. (**A**) Left panel shows representative 100× images of phospho-Smad 1/5/8 staining in mutant *APP*-infected human fetal neurospheres compared to Zsgreen controls. Right panel shows quantification of DAPI and phospho-Smad1/5/8 co-stained cells in each group. Data are presented as mean ± SD. (**B**) Representative 100× images of phospho-Smad1/5/8 staining in neurospheres treated with LDN-193189 for one week. (**C**) Quantification of phospho-SMAD 1/5/8 after treatment with different doses of LDN-193189. A two-way ANOVA revealed significant differences between the groups (**** for p<0.0001). Data are presented as mean ± SD. (**D**) Representative 6× images of in vitro colonies of mutant *APP*- and Zsgreen-infected human fetal neurospheres after one week of LDN-193189 treatment. (**E**) Quantification of the colonies in (**D**). A two-way ANOVA revealed significant differences between groups (**** for p<0.0001 and *** for p=0.0003). Data are presented as mean ± SD.

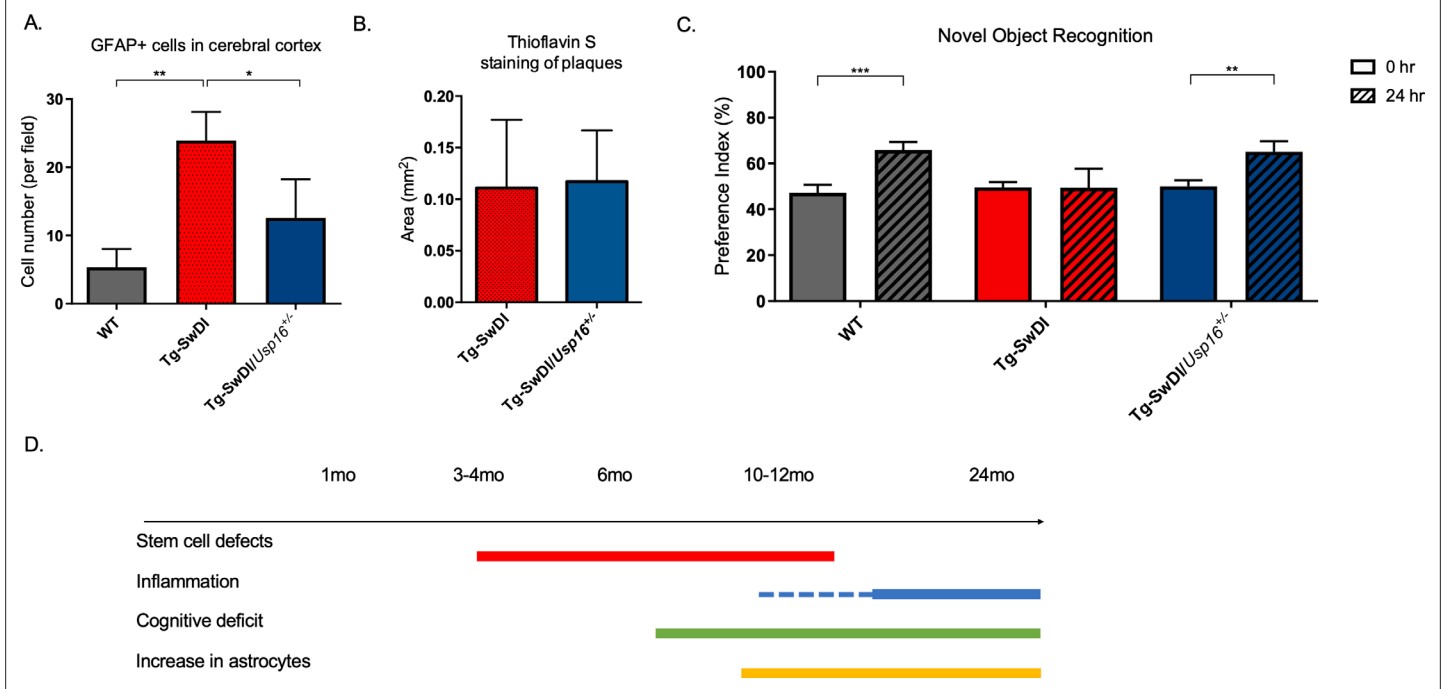

**Figure 7.** Astrogliosis, cognitive deficits, but not amyloid plaque burden are some of the processes rescued in Tg-SwDI/*Usp16*⁺/⁻ mice. (**A**) Anterior sections were obtained from 9 to 12 months old mice, stained, and counted for GFAP+ cells in the cortex. Four different images per sections and three sections per mouse were counted (n = 4). Bar graph shows quantification of GFAP+ cells from cortex. A one-way ANOVA showed significant differences between the groups (p=0.0012 between wild type (WT) and Tg-SwDI and p=0.0188 between Tg-SwDI and Tg-SwDI/*Usp16*⁺/⁻). Data are presented as mean ± SD. See also ***Figure 7—figure supplement 1A***. (**B**) Representative images of thioflavin S on the left. On the right, quantification of area covered by plaques using thioflavin S staining in Tg-SwDI and Tg-SwDI/*Usp16*⁺/⁻ mice shows no difference between the two genotypes (10-month-old mice). Data are presented as mean ± SEM. See also ***Figure 7—figure supplement 1B***. (**C**) Novel object recognition 24 hr testing in mice at 6 months of age showed the earliest signs of cognitive impairment in the Tg-SwDI mice with a preference index (PI) of 49%, while WT and Tg-SwDI/*Usp16*⁺/⁻ mice had PIs >65% indicating intact object discrimination (p=0.001 for WT and p=0.0099 for Tg-SwDI/*Usp16*⁺/⁻, n = 7–10 mice). Data are presented as mean ± SEM. See also ***Figure 7—figure supplement 3***. (**D**) Schematic summarizing the temporal effects of mutant *APP* demonstrated in this manuscript.

The online version of this article includes the following figure supplement(s) for figure 7:

**Figure supplement 1.** *Usp16* haploinsufficiency partially rescues GFAP+ cells in the Tg-SwDI cortex.

**Figure supplement 2.** No change in inflammation is present in Tg-SwDI or Tg-SwDI/*Usp16*⁺/⁻ mice at 1-year old.

**Figure supplement 3.** Long-term and spatial memory was improved with *Usp16* haploinsufficiency in Tg-SwDI mice.

(p<0.0001, ***Figure 6B C***; *Yu et al., 2008*). Furthermore, when we treated neurospheres expressing mutant *APP* with LDN-193189 for a week, the number of colonies originating from those cells were similar to control cells and significantly higher than untreated mutant *APP* neurospheres (***Figure 6D E***). Notably, LDN-193189 had minimal impact on Zsgreen control neurosphere growth (***Figure 6E***). This finding demonstrates that the decrease in NIC frequency observed with mutant *APP* could be explained in part by the upregulation of BMP signaling. Moreover, BMPR inhibition rescues this defect in cells overexpressing mutant *APP* at doses that have minimal toxicity on healthy cells. Altogether, these data reveal that BMP signaling enrichment is recapitulated in human NPCs expressing mutant *APP*, and that BMPR inhibition normalizes the stem cell defect.

## Astrogliosis is reduced and cognitive function is restored in Tg-SwDI/*Usp16*⁺/⁻ mice

Having identified USP16 as a target to modulate two critical pathways affected by mutations in *APP*, *Cdkn2a*, and BMP, we further investigated USP16's potential effects on downstream pathophysiological markers of AD that are recapitulated in the Tg-SwDI model such as astrogliosis, inflammation, amyloid plaques and memory. Increased numbers of GFAP+ astrocytes were seen throughout the cortex of 9–12-month-old Tg-SwDI mice, which could mark the beginning of astrogliosis, and were

significantly reduced with *Usp16* haploinsufficiency (*Figure 7A*, *Figure 7—figure supplement 1A*, see also *Figure 3—figure supplement 2*).

Amyloid plaques are one of the defining features of AD, and controversy exists concerning the effect of plaques on cognitive decline. Mutations in *APP* lead to amyloid plaque deposition throughout the brain as seen in 10-month-old Tg-SwDI mice (*Figure 7B*, *Figure 7—figure supplement 1B*). However, no difference was observed in plaque burden, demonstrated by Thioflavin S staining, in the age-matched Tg-SwDI/*Usp16*+/- mice (*Figure 7B*, *Figure 7—figure supplement 1B*). In addition, a Luminex screen of 1-year-old Tg-SwDI/*Usp16*+/- mice also did not reveal significant differences in the levels of inflammatory cytokines from any of the groups (*Figure 7—figure supplement 2A–C*).

As expected, when studying the cognitive decline in the Tg-SwDI cohort, we found that the Tg-SwDI cohort exhibited impaired performance in the NOR task as early as 6 months of age, with preference indexes (PIs) that were not significantly different 24 hr after training, indicating no memory of the familiar object (*Figure 7C*). The Tg-SwDI/*Usp16*+/- mice performed equally to their age-matched WT controls indicating memory of the familiar object with PIs in the 65–70% range (p=0.001 and p=0.0099, respectively; *Figure 7C*). Long-term memory impairment in Tg-SwDI mice and rescue in Tg-SwDI/*Usp16*+/- mice was further supported by the Barnes maze (BM) where Tg-SwDI mice spent more time exploring off-target quadrants and Tg-SwDI/*Usp16*+/- mice spent more time in the target quadrant (p=0.0128 and p=0.0251, respectively; *Figure 7—figure supplement 3*). These data indicate that although modulating *Usp16* gene dosage does not affect amyloid plaque burden, it ameliorates stem cell self-renewal defects that may be the earliest indication of pathology, as well as some of the cognitive defects in these mice that occur later (*Figure 7D*).

## Discussion

Numerous studies have sought to target processes such as inflammation, amyloid plaque accumulation, and ROS to AD pathologies in both humans and mouse models (*Gjoneska et al., 2015*; *Sevigny et al., 2016*). The lack of robust efficacy in trials that utilize therapies directed against amyloid and inflammatory pathways, even when initiated relatively early in the disease (*Selkoe, 2019*; *Imbimbo et al., 2010*), suggests that other mechanisms are at play. If so, identification of these other disease mechanisms is needed to develop effective treatments (*Aisen, 2008*; *Doody et al., 2013*; *Green et al., 2009*; *Group et al., 2008*; *Salloway et al., 2014*).

One of the primary findings in this study is that an NPC defect predates the development of measurable inflammation and amyloid plaque accumulation in a mutant *APP* model and that this defect is cell intrinsic. We further show this cell intrinsic NPC defect is reproduced in human fetal NPCs expressing *APP* Swedish and Indiana mutations, suggesting that our findings are translatable to other *APP* mutations and to human cells. The NPC defect that we discovered is partly regulated by *Cdkn2a*, a central component of aging responsible for decreased neurogenesis and differentiation of NPCs (*Abdouh et al., 2012*; *Molofsky et al., 2006*). Because *CDKN2A* expression has been correlated with sporadic AD (*Arendt et al., 1996*; *Lüth et al., 2000*; *McShea et al., 1997*) and targeting *CDKN2A* has shown benefit in improving age related and sporadic AD in models, we speculate our findings may have relevance for treatment of sporadic AD.

Current strategies to reverse neurogenesis defects include the use of drugs ('senolytics') that selectively remove p16$^{Ink4a}$-positive senescent cells. Removal of the p16$^{Ink4a}$-positive senescent cells, for instance, using a suicide gene under the regulation of the *Cdkn2a* promoter has been shown to attenuate progression of age-related decline and preserve cognitive function in both an accelerated aging AD mouse model and a tauopathy mouse model (*Baker et al., 2011*; *Bussian et al., 2018*). However, the use of a suicide gene is not directly translatable into humans, and other senolytics such as BCL2-inhibitors or the combination of Dasatinib and quercetin have toxicities which can limit their use (*Amaya-Montoya et al., 2020*; *Zhu et al., 2015*).

Here, we propose USP16 as a novel target that might circumvent many of the problems associated with *CDKN2A* inhibition or with senolytics' treatment. When we inhibited USP16 by making Tg-SwDI mice haploinsufficient for *Usp16* (Tg-SwDI/*Usp16*+/-), we found a rescue in the self-renewal of NPCs as early as 3 months of age. We also demonstrated a new role for USP16 in regulating the BMP pathway, a mechanism independent of *Cdkn2a*. Previously, Gargiulo et al. found that self-renewal gene *Bmi1*, whose PRC1 activity is counterbalanced by USP16, represses BMP signaling (*Gargiulo et al., 2013*). NPCs from *Bmi1* knockout mice treated with BMP4 experience even further growth arrest than those

untreated (*Gargiulo et al., 2013*). Furthermore, Kwak, Lohuizen, and colleagues showed that treatment of human neural stem cells with secreted APPα or overexpression of *APP* promoted phosphorylation of SMAD 1/5/8 and induced massive glial differentiation (by expression of GFAP) through the BMP pathway (*Kwak et al., 2014*). In line with this, our results reveal expression of mutant *APP* in human fetal NPCs induced phosphorylation of SMAD 1/5/8 and reduced neurosphere colony formation that was rescued by a BMP receptor inhibitor. Importantly, our data extends their findings of astrogliosis to an in vivo mouse model of AD. Interestingly, BMI1 regulates both *Cdkn2a* and the BMP pathway independently, and BMI1 expression was shown to be decreased in AD patients compared to age-matched controls (*Flamier et al., 2018*).

The timing of therapeutic treatment in AD seems to be crucial as studies have shown that treating the disease too late has little efficacy (*Sperling et al., 2011*; *Yiannopoulou et al., 2019*). The Tg-SwDI mice used in our study develop cognitive defects after the accumulation of amyloid beta plaques, a timeline which mimics that of humans with both dominantly inherited AD and late-onset sporadic AD (*Bateman et al., 2012*). In a study characterizing sporadic AD, for instance, Villemagne et al. used Pittsburgh compound B (PiB) positron emission tomography (PET) to show that 31% of healthy control subjects had high PiB retention indicating Aß deposition and of these, 25% developed mild cognitive impairment or AD by 3 years (*Villemagne et al., 2011*). Furthermore, studies such as those by Salloway et al. and Biogen's EMERGE and ENGAGE trials found that treating the plaques alone did not rescue cognitive defects (*Knopman et al., 2021*; *Salloway et al., 2014*). Although the mechanism for how mutant *APP* causes an NPC defect is outside the scope of this work, it has been postulated that amyloid-ß oligomers may impair neurogenesis and promote gliogenesis of human NPCs through the GSK-3ß pathway which unfortunately is not amenable to inhibition as it phosphorylates a variety of substrates and incurs large cytotoxic off-target effects (*Bernabeu-Zornoza et al., 2019*; *Lee et al., 2013*). The translatability of our study therefore comes from observing that early therapeutic reduction of USP16 or BMP signaling in neural stem cells may reverse the neurogenic defect that may contribute to symptomatic AD later in life, especially if applied before cognitive deficits are present in the patient.

Understanding the pathophysiology of a disease is critical to developing therapeutic targets and designing intervening therapies. Here, we present USP16 as a potential therapeutic target acting on both BMP and *Cdkn2a* pathways independently. It is important to note that USP16 reduction also reduced astrocyte proliferation and restored cognitive function as measured by the NOR and Barnes Maze tests, independently of plaques and widespread inflammation. The increase in GFAP-expressing cells and impaired cognitive function seen in this AD model are purely attributable to mutant *APP* as Tg-SwDI mice do not develop neurofibrillary tangles that require mutations in tau (*Wilcock et al., 2008*). Thus, therapeutic strategies that combine targeting USP16, which effectively rescue the mutant *APP*-induced cell intrinsic damage, with agents that target extracellular plaque formation, neurofibrillary tangles, and/or inflammation may improve treatments for AD.

## Materials and methods

**Key resources table**

| Reagent type (species) or resource | Designation | Source or reference | Identifiers | Additional information |
|---|---|---|---|---|
| Gene (*Homo sapiens*) | Tg-SwDI | *Davis et al., 2004* | *APP* KM670/671NL (Swedish), *APP* E693Q (Dutch), *APP* D694N (Iowa) | Transgenic mutant *APP* in mouse model with Swedish, Dutch, and Iowa mutations |
| Gene (*Homo sapiens*) | *APP* SwI | *Young-Pearse et al., 2007* | *APP* K595N (Swedish), *APP* M596L and V642F (Indiana) | Mutant human *APP* in human neurosphere model with Swedish and Indiana mutations |
| Strain, strain background (*Mus musculus*) | C57BL/6-Tg(*Thy1-APP*SwDutIowa)BWevn/Mmjax, C57Bl/6 | The Jackson Laboratory | Jax Stock#: 007027; RRID:MMRRC_034843-JAX | Tg-SwDI mice |
| strain, strain background (*Mus musculus*) | FVB/N-*Usp16*<sup>Tg(Tyr)2414FOve</sup>/Mmjax, FVB | Mutant Mouse Regional Resource Centers | Jax Stock#: 036225-JAX; RRID:MMRRC_036225-JAX | *Usp16*<sup>+/-</sup> mice |
| Strain, strain background (*Mus musculus*) | B6.129-*Cdkn2a*<sup>tm1Rdp</sup>/Nci, B6.129 | Mouse Models of Human Cancers Consortium | MMHCC strain:#01XB1; RRID:IMRS_NCIMR:01XB1 | *Cdkn2a*<sup>-/-</sup> mice |
| Transfected construct (*Homo sapiens*) | pHIV-ZsGreen | Addgene | RRID:Addgene_18121 | Empty Lentiviral Backbone as control |

*Continued on next page*

*Continued*

| Reagent type (species) or resource | Designation | Source or reference | Identifiers | Additional information |
|---|---|---|---|---|
| Transfected construct (*Homo sapiens*) | *APP* SwI | This paper (cloned) | pHIV-*APP*SwI | To model mutant *APP* neurospheres |
| Transfected construct (*Homo sapiens*) | *APP* WT | This paper (cloned) | pHIV-*APP*695 | To model WT *APP* neurospheres |
| Biological sample (*Mus musculus*) | Primary neural stem cells | The Jackson Laboratory | Tg-SwDI, WT, Tg-SwDI/*Usp16*$^{+/-}$, *Usp16*$^{+/-}$, Tg-SwDI/*Cdkn2a*$^{-/-}$, *Cdkn2a*$^{-/-}$ | Freshly isolated from *M. musculus* |
| Biological sample (*Homo sapiens*) | Primary human fetal neural stem cells | University of California Irvine | | Isolated from 18 week fetal neural tissue, enriched for CD133+ cells |
| Antibody | anti-SOX2 (Goat polyclonal) | R&D Systems | Cat# AF2018, RRID:AB_355110 | IHC(1:50) |
| Antibody | Anti-GFAP (Rabbit polyclonal) | Stem Cell Technologies | Cat#:60128, RRID:AB_1118515 | IHC(1:500) |
| Antibody | Anti-pSMAD1/5/8 (Rabbit monoclonal) | CST | Cat#:9516, RRID:AB_491015 | IF(1:100) |
| Antibody | Anti-beta-amyloid (Mouse monoclonal) | Invitrogen | Cat#:13–0200, RRID:AB_2532993 | IF(1:100) |
| antibody | Pacific Blue anti-mouse CD31 Antibody (Rat monoclonal) | Biolegend | Cat#:102421, RRID:AB_10613457 | FACS(5 µl per test) |
| Antibody | Pacific Blue anti-mouse CD45 Antibody (Rat monoclonal) | Biolegend | Cat#:103125, RRID:AB_493536 | FACS(5 µl per test) |
| Antibody | Pacific Blue anti-mouse TER-119 Antibody (Rat monoclonal) | Biolegend | Cat#116231, RRID:AB_2149212 | FACS(5 µl per test) |
| Antibody | FITC anti-mouse CD24 Antibody (Rat monoclonal) | Biolegend | Cat#101805, RRID:AB_312838 | FACS(5 µl per test) |
| Recombinant DNA reagent | pCAX *APP* Swe/Ind (plasmid) | Addgene | RRID:Addgene_30145 | Mutant *APP* with Swedish, Indiana mutations |
| Recombinant DNA reagent | pCAX *APP* 695 | Addgene | RRID:Addgene_30137 | Wild type *APP* |
| Recombinant DNA reagent | pHIV-Zsgreen (plasmid) | Addgene | RRID:Addgene_18121 | Lentiviral backbone |
| Commercial assay or kit | RNeasy Lipid Tissue Kit | Qiagen | Cat#: 74,804 | |
| Commercial assay or kit | Click-iT EdU cell proliferation kit | Invitrogen | Cat#: C10337 | |
| Commercial assay or kit | Nextera XT Library Sample Preparation Kit | Illumina | Cat#: FC-131–1096 | |
| Chemical compound, drug | LDN-193189 | Sigma Aldrich | S2618 | 5 nM and 50 nM |
| Software, algorithm | R | R | RRID:SCR_001905 | Single cell RNA-seq |
| Software, algorithm | GSEA | http://www.broadinstitute.org/gsea/ | RRID:SCR_003199 | Gene set enrichment analysis |
| Software, algorithm | ELDA | http://bioinf.wehi.edu.au/software/elda/ | RRID:SCR_018933 | Limiting dilution experiments |
| Software, algorithm | ImageJ | https://imagej.net/ | RRID:SCR_003070 | IF analysis |
| Software, algorithm | Transcriptome Analysis Console | Thermo Fisher | RRID:SCR_016519 | Microarray analysis |
| Other | DAPI stain | Sigma | 32,670 | IHC (1:10000) |
| Other | Thioflavin S | Sigma | 1326-12-1 | IHC (1%) |
| Other | Sytox Blue | Thermo Fisher | S11348 | FACS |

## Statistical analyses

In all the graphs, bars show average as central values and ±SD as error bars, unless otherwise specified. *P* values were calculated using ANOVA in analyses with three or more groups. Tukey's method was used for multiple test correction with 95% simultaneous confidence levels. Two-tailed *t*-tests were used in analyses comparing two groups, unless otherwise specified. For limiting dilution analyses, ELDA software was used to test inequality between multiple groups. Expected frequencies are reported, as well as the 95% confidence intervals (lower and upper values are indicated). *p<0.05, **p<0.01, ***p<0.001.

## Mice

Tg-SwDI mice (background C57Bl/6) were purchased from Jackson Laboratories and housed in cages of 5 mice. These mice were made hemizygous for experiments after breeding with *Cdkn2a*$^{-/-}$ (C57Bl6

background) or $Usp16^{+/-}$ mice (back-crossed to B6EiC3). $Usp16^{+/-}$ mice were originally ordered from Mutant Mouse Regional Resource Centers (MMRRC) and $Cdkn2a^{-/-}$(B6.129-$Cdkn2a^{tm1Rdp}$) were obtained from Mouse Models of Human Cancers Consortium (NCI-Frederick). WT littermates were used as control mice. Mice were maintained in cages of 5 and genotyped by traditional PCR according to animal's provider. Mice were housed in accordance with the guidelines of Institutional Animal Care Use Committee. All animal procedures and behavioral studies involved in this manuscript are compliant to Stanford Administrative Panel on Laboratory Animal Care Protocol 10,868 pre-approved by the Stanford Institutional Animal Care and Use Committee.

## Immunohistochemistry

All animals were anesthetized with avertin and transcardially perfused with 15 ml phosphate-buffered saline (PBS). Brains were postfixed in 4% paraformaldehyde overnight at 4°C before cryo-protection in 30% sucrose. Brains were embedded in optimum cutting temperature (Tissue-Tek) and coronally sectioned at 40 µm using a sliding microtome (Leica, HM450). For immunohisto-chemistry, sections were stained using the Click-iT EdU cell proliferation kit and protocol (Invit-rogen) to expose EdU labeling followed by incubation in blocking solution [3% normal donkey serum, 0.3% Triton X-100 in PBS] at room temperature for 1 hr. Goat antibody to Sox2 (anti-Sox2) (1:50; R&D Systems AF2018) and rabbit anti-GFAP (1:500; Stem Cell Technologies 60128) were diluted in 1% blocking solution (normal donkey serum in 0.3% Triton X-100 in PBS) and incu-bated overnight at 4°C. Secondary-only stains were performed as negative controls. The following day, sections were rinsed three times in ×1 PBS and incubated in secondary antibody solution (1:500) and 4',6-diamidino-2-phenylindole (DAPI) (1:10,000) in 1% blocking solution at 4°C for 4 hr. The following secondary antibodies were used: Alexa 594 donkey anti-rabbit (Jackson Immu-noResearch), Alexa 647 donkey anti-goat (Jackson ImmunoResearch). The next day, sections were rinsed three times in PBS and mounted with ProLong Gold Antifade (Cell Signaling) mounting medium. For senile plaques, sections were incubated for 8 min in aqueous 1% Thioflavin S (Sigma) at room temperature, washed in ethanol and mounted. Total plaque area from images taken of 6 sections (1 technical replicate = 1 section) were analyzed from each mouse with n = 3 mice (1 biological replicate = 1 mouse) in each group.

## Confocal imaging and quantification

All cell counting was performed by experimenters blinded to the experimental conditions using a Zeiss LSM700 scanning confocal microscope (Carl Zeiss). For EdU stereology, all EdU-labeled cells in every 6th coronal section of the SVZ were counted by blinded experimenters at ×40 magnification. The total number of EdU-labeled cells co-labeled with SOX2 and GFAP per SVZ was determined by multiplying the number of $EdU^+GFAP^+SOX2^+$ cells by 6. Cells were considered triple-labeled when they colocalized within the same plane.

## Mouse neurosphere cultures

To produce neurospheres, mice were euthanized by $CO_2$, decapitated and the brain immediately removed. The subventricular zone was micro-dissected and stored in ice-cold PBS for further processing. The tissue was digested using Liberase DH (Roche) and DNAse I (250 U/ml) at 37°C for 20 min followed by trituration. Digested tissue was washed in ice-cold HBSS without calcium and magnesium, filtered through a 40 µm filter and immediately put into neurosphere growth media that is, Neurobasal-A (Invitrogen) supplemented with Glutamax (Life Technologies), 2% B27-A (Invi-trogen), mouse recombinant epidermal growth factor (EGF; 20 ng/ml) and basic fibroblast growth factor (bFGF; 20 ng/ml) (Shenandoah Biotechnology).

For limiting dilution analysis, cells were directly plated into 96-well ultra-low adherent plates (Corning Costar) in limiting dilutions down to one cell per well. Each plating dose was done in tech-nical replicates of up to 12 wells in each experiment, and the number of wells with neurospheres was counted after 10 days. For passaging, neurospheres were dissociated and re-plated at a density of 10 cells/µl. Experiment was repeated three times (each infection and subsequent limiting dilution exper-iment performed being a biological replicate).

## RNA expression analyses (mouse)

For gene expression analyses, cells were collected in Trizol (Invitrogen), and RNA was extracted following the manufacturer's protocol. Complementary DNA was obtained using Superscript III First Strand Synthesis (Invitrogen). Real-time PCR reactions were assembled using Taqman probes (Applied Biosystems) in accordance with the manufacturer's directions. Expression data were normalized by the expression of housekeeping gene *Actb* (Mm00607939_s1). Probes used in this study: *Cdkn2a* (Mm_00494449), *Bmi1* (Mm03053308_g1), *Il1b* (Mm01336189_m1), *Il6* (Mm99999064_m1), *Tnf* (Mm00443258_m1), *Cox2* (Mm03294838_g1), *Aspg* (Mm01339695_m1), *C3* (Mm01232779_m1), *Cd14* (Mm00438094_g1), *Cd44* (Mm01277160_m1), *Clcf1* (Mm01236492_m1), *Emp1* (Mm00515678_m1), *Gfap* (Mm01253033_m1), *Ggta1* (Mm01333302_m1), *S1pr3* (Mm00515669_m1), *Serping1* (Mm00437835_m1), *Slc10a6* (Mm00512730_m1), *Srgn* (Mm01169070_m1), *Stat3* (Mm01219775_m1), *Vim* (Mm00449201_m1). Biological replicates of a minimum of 3 mice and N = 2 technical replicates for each mouse were used.

## Microarray

SVZ, DG, and Cortex were dissected from 2-year-old mice, homogenized using Qiagen TissueRupture, and RNA was extracted using RNeasy Lipid Tissue Kit (Qiagen). RNA was submitted to Stanford PAN facility where amplification of cDNA and hybridization to Mouse Gene 2.0 ST array was performed. Microarray analyses were carried out using the TAC (Transcriptome Analysis Console) from Thermofisher. TAC includes the normalization, probe summarization, and data quality control functions of Expression Console Software. The expression analysis settings were set as fold change <-2 or >2 with a p-value <0.05 using ebayes ANOVA method. The heat map clustering was generated using a gene list including the differentially expressed genes between WT and Tg-SwDI with a conditional *F*-test <0.05.

## Brain multianalyte analysis

The different brain regions were lysed using cell lysis buffer (Cell signaling #9803) with PMSF (Cell signaling #8553) and complete mini EDTA free protease inhibitor followed by mechanical homogenation by Tissue Ruptor (Qiagen). The samples were centrifuged at 13,000 rpm for 15 min and protein concentration calculated by BCA. Normalized samples were analyzed by the Stanford Human Immune Monitoring Center using a Luminex mouse 38-plex analyte platform that screens 38 secreted proteins using a multiplex fluorescent immunoassay. Brain homogenates were run in technical duplicates (2 wells with 200 µg each from each biology replicate) with three biological replicates (1 biological replicate = 1 brain from 1 mouse). The Luminex data (mean RFI) was generated by taking the raw fluorescence intensities of each sample and dividing by a control sample (one of the WT samples), then taking the average of the triplicated samples for each genotype.

## Behavioral testing
### Novel object recognition

One behavioral test used in this study for assessing long term memory was NOR [67] carried out in arenas (50cm × 50 cm × 50 cm) resting on an infra-red emitting base. Behavior was recorded by an infrared-sensitive camera placed 2.5 m above the arena. Data were stored and analyzed using Videotrack software from ViewPoint Life Sciences, Inc (Montreal, Canada) allowing the tracking of body trajectory/speed and the detection of the nose position. On the day before NOR training, the mouse was habituated to the apparatus by freely exploring the open arena. NOR is based on the preference of mice for a novel object versus a familiar object when allowed to explore freely. For NOR training, two identical objects were placed into the arena and the animals were allowed to explore for 10 min. Testing occurred 24 hr later in the same arena but one of the familiar objects used during training was replaced by a novel object of similar dimensions, and the animal was allowed to explore freely for 7 min. The objects and the arena were cleaned with 10% ethanol between trials. Exploration of the objects was defined by the time spent with the nose in a 2.5 cm zone around the objects. The PI was calculated as the ratio of the time spent exploring the novel object over the total time spent exploring the two objects. The PI was calculated for each animal and averaged among the groups of mice by genotype. The PI should not be significantly different from 50% in the training session, but is significantly different if novelty is detected.

## Barnes maze

Another test of long-term memory that is indicative of spatial memory is the Barnes Maze similar to that described by *Attar et al., 2013*. The Barnes maze is a 20-hole circular platform measuring 48" in diameter with holes cut 1.75" in diameter and 1" from the edge. The platform is elevated 100 cm above the floor, and is located in the center of a room with many extra-maze and intra-maze visual cues. This task takes advantage of the natural preference of rodents for a dark environment. Motivated to escape the bright lights and the open-space of the platform, rodents search for an escape hole that leads to a dark box beneath the maze and with training they learn to use distal visual cues to determine the spatial location of the escape hole. A habituation day was followed by training over 2 days and a test day separated by 24 hr. Two downward-facing 150-watt incandescent light bulbs mounted overhead served as an aversive stimulus. Mice completed three phases of testing: habituation, training, and the probe test.

For habituation, mice were placed within the start cylinder in the middle of the maze to ensure random orientation for 15 s. The overhead lights were then turned on and mice were given 3 min to independently enter through the target hole into the escape cage. If a mouse did not enter the escape box freely, the experimenter coaxed the mouse to enter the escape box by touching the mouse's tail.

For training, a mouse was placed in the middle of the maze in random orientation for 15 s. The overhead lights were turned on, and the tracking software was activated. The mouse was allowed up to 3 min to explore the maze and enter the escape hole. If it failed to enter within 3 min, it was gently guided to the escape hole using the start cylinder and allowed to enter the escape cage independently.

On the test/probe day, 24 hr after the last training day, the experiment was set up as described on training days, except the target hole was covered. The percent time in the correct zone and average proximity to the correct escape hole are more sensitive measures of memory than percentage visits to the correct hole. Therefore, during the probe phase, measures of time spent per quadrant and holes searched per quadrant were recorded. For these analyses, the maze was divided into quadrants consisting of five holes with the target hole in the center of the target quadrant. On day 4, latency (seconds) and path length (meters) to reach the target hole were measured. Number of pokes in each hole were calculated, time spent per quadrant and holes searched per quadrant were recorded and paired *t*-tests were used to compare the percentage of time spent between quadrants.

## Differentiation

Neurospheres derived from the SVZ of 1-year-old mice were dissociated into single cells and 2000 cells were cultured per well on PDL and laminin-coated adherent 96-well cell culture plates (mouse neurospheres used for differentiation). The cells were cultured in Neurobasal-A (Invitrogen) media containing 1% fetal bovine serum. After 6 days in culture, the cells were stained directly in the wells using the Stemcell Technologies protocol. Wells were incubated for 2 hr at room temperature in primary rabbit antibody to GFAP (1:200, Dako Z0334) followed by three washes in ×1 PBS and incubated in secondary antibody solution Alexa-647 goat anti-rabbit (1:500; Jackson ImmunoResearch) and 4',6-diamidino-2-phenylindole (DAPI) (1:10,000). Cells co-positive for GFAP and DAPI were counted using ImageJ and divided by total number of DAPI-positive cells. Experiment was performed with three biological replicates in triplicate (three technical replicates and three images were taken of differing regions of each technical replicate/well).

## Human neurosphere cultures

A human fetal neural stem cell line from University of California Irvine was developed from fetal neural tissue at eighteen-week gestational age enriched for CD133+ cells. The cells were negative for mycoplasma and viral contaminants using qPCR (IDEXX BioResearch) and had normal karyotype. The use of neural progenitor cells as non-hESC stem cells in this study is compliant to Stanford Stem Cell Research Oversight (SCRO) Protocol 194 pre-approved by the Internal Review Board (IRB)/SCRO of the Stanford Research Compliance Office (RCO). Informed consent was obtained, and standard material transfer agreement signed. Cells were grown in nonadherent ultra-low attachment well plates in X-VIVO 15 media (LONZA) supplemented with LIF (10 ng/ml), N2 Supplement, N-acetylcysteine (63 ug/ml), Heparin (2 ug/ml), EGF (20 ng/ml), and FGF (20 ng/ml).

For limiting dilution analysis, cells were directly plated into 96-well ultra-low adherent plates (Corning Costar) in limiting dilutions down to one cell per well. Each plating dose was done in technical replicates of up to 12 wells in each experiment, and the number of wells with neurospheres was counted after 10 days. Experiment was repeated three times (each infection and subsequent limiting dilution experiment performed being a biological replicate).

### Lentivirus production

pCAX *APP* Swe/Ind was a gift from Dennis Selkoe & Tracy Young-Pearse (Addgene plasmid #30145; http://n2t.net/addgene:30145; RRID:Addgene_30145). pCAX *APP* 695 was a gift from Dennis Selkoe & Tracy Young-Pearse (Addgene plasmid #30137; http://n2t.net/addgene:30137; RRID:Addgene_30137). pHIV-Zsgreen was a gift from Bryan Welm & Zena Werb (Addgene plasmid #18121; http://n2t.net/addgene:18121; RRID:Addgene_18121). The cDNA for mutant *APP* harboring the Swedish and Indiana mutations or the cDNA for wild type *APP* from the plasmids listed above were each cloned into a pHIV-Zsgreen backbone obtained from Addgene (also listed above). Lipofectamine 2000 was used to transduce the construct (either pHIV-Zsgreen+mutant *APP* or pHIV-Zsgreen alone) into H293T cells and media was collected after 48 hr. Virus was ultra-centrifuged and resuspended in PBS then titered before infecting human fetal neurospheres.

### Flow cytometry

For single-cell RNA-sequencing, the subventricular zone of 4 mice from each genotype was micro-dissected and tissue digested using Liberase DH (Roche) and DNAse I (250 U/ml) at 37°C for 20 min followed by trituration. Digested tissue was washed in ice-cold HBSS without calcium and magnesium, filtered through a 40 µm filter, and then stained with the following antibodies for 30 min: PacBlue-CD31 (Biolegend), PacBlue-CD45 (Biolegend), PacBlue-TER119 (Biolegend), and FITC-CD24 (Biolegend). Sytox Blue was used for cell death exclusion and samples were sorted into 384 well plates prepared with lysis buffer using the Sony Sorter.

### Lysis plate preparation

Lysis plates were created by dispensing 0.4 µl lysis buffer (0.5 U Recombinant RNase Inhibitor (Takara Bio, 2,313B), 0.0625% Triton X-100 (Sigma, 93443–100 ML), 3.125 mM dNTP mix (Thermo Fisher, R0193), 3.125 µM Oligo-dT30VN (IDT, 5′-AGCAGTGGTATCAACGCAGAGTACT30VN-3′) and 1:600,000 ERCC RNA spike-in mix (Thermo Fisher, 4456740)) into 384-well hard-shell PCR plates (Biorad HSP3901) using a Tempest or Mantis liquid handler (Formulatrix).

### cDNA synthesis and library preparation

cDNA synthesis was performed using the Smart-seq2 protocol [1,2]. Illumina sequencing libraries were prepared according to the protocol in the Nextera XT Library Sample Preparation kit (Illumina, FC-131–1096). Each well was mixed with 0.8 µl Nextera tagmentation DNA buffer (Illumina) and 0.4 µl Tn5 enzyme (Illumina), then incubated at 55°C for 10 min. The reaction was stopped by adding 0.4 µl 'Neutralize Tagment Buffer' (Illumina) and spinning at room temperature in a centrifuge at 3220 × *g* for 5 min. Indexing PCR reactions were performed by adding 0.4 µl of 5 µM i5 indexing primer, 0.4 µl of 5 µM i7 indexing primer, and 1.2 µl of Nextera NPM mix (Illumina). PCR amplification was carried out on a ProFlex 2 × 384 thermal cycler using the following program: 1. 72°C for 3 min, 2. 95°C for 30 s, 3. 12 cycles of 95°C for 10 s, 55°C for 30 s, and 72°C for 1 min, and 4. 72°C for 5 min.

### Library pooling, quality control, and sequencing

Following library preparation, wells of each library plate were pooled using a Mosquito liquid handler (TTP Labtech). Pooling was followed by two purifications using ×0.7 AMPure beads (Fisher, A63881). Library quality was assessed using capillary electrophoresis on a Fragment Analyzer (AATI), and libraries were quantified by qPCR (Kapa Biosystems, KK4923) on a CFX96 Touch Real-Time PCR Detection System (Biorad). Plate pools were normalized to 2 nM and equal volumes from 10 or 20 plates were mixed together to make the sequencing sample pool. PhiX control library was spiked in at 0.2% before sequencing. Single-cell libraries were sequenced on the NovaSeq 6000 Sequencing System (Illumina) using 2 × 100 bp paired-end reads and 2 × 8 bp or 2 × 12 bp index reads with a 300-cycle kit (Illumina 20012860).

## Data processing

Sequences were collected from the sequencer and de-multiplexed using bcl2fastq version 2.19.0.316. Reads were aligned using to the mm10plus genome using STAR version 2.5.2b with parameters TK. Gene counts were produced using HTSEQ version 0.6.1p1 with default parameters, except 'stranded' was set to 'false,' and 'mode' was set to 'intersection-nonempty.' As mentioned above, four biological replicates from each genotype and at each age were combined for the single-cell RNA-seq experiment (16 samples per age group). Basic filtering of cells and genes was conducted pre-analysis using the Seurat package in R (*Butler et al., 2018*). Briefly, genes that were not expressed in a minimum of 5 cells were filtered out, and cells had to have a minimum of 50,000 reads and a maximum of 3,000,000 reads. Similarly, cells with less than 500 or more than 5000 genes were filtered out. This left us with the following numbers of cells below after filtering. In this experiment, a biological replicate is defined as 1 mouse of a specific genotype and a technical replicate is defined as one cell.

|  | 3 and 4 months | 1-year- old |
|---|---|---|
| WT | 1008 | 980 |
| Tg-SwDI | 642 | 1089 |
| Tg-SwDI/*Usp16*+/- | 712 | 729 |
| *Usp16*+/- | 651 | 731 |

## Gene set enrichment analysis

Gene counts were log normalized and scaled before generating the.gct files. GSEA with the Hallmarks gene sets was run with standard parameters: 1000 permutations of type phenotype, with no collapsing to gene symbols, and weighted enrichment. Gene sets were considered significantly enriched if FDR < 25%.

## Immunofluorescence (human neurospheres)

Neurospheres were cytospun onto slides and fixed in ice-cold methanol for 5 min. Slides were rinsed three times in PBS at room temperature, followed by blocking in 3% BSA in PBS for 1 hr at room temperature. Rabbit antibody to pSMAD 1/5/8 (1:100; CST 9516) and mouse antibody to beta-amyloid (1:100; Invitrogen 13–200) were diluted in the same 3% blocking buffer and incubated overnight at 4°C. The following day, sections were rinsed three times in ×1 PBS and incubated in secondary antibody solution Cy-3 donkey anti-rabbit (1:500; Jackson ImmunoResearch) or Cy-3 donkey anti-mouse (1:500; Jackson ImmunoResearch) and 4′,6-diamidino-2-phenylindole (DAPI) (1:10,000) in 3% blocking solution at room temperature for 2 hr. Slides were then washed 3 times at room temperature in ×1 PBS and mounted. Cells positive for pSMAD 1/5/8 were counted by ImageJ. Experiment was performed three times (1 biological replicate = 1 round of infection with subsequent experiment) in triplicate (1 technical replicate = 1 slide with at least 3 neurospheres with 1 neurosphere having at least 100 cells).

## Colony counts

Human neurospheres were dissociated into single cells and infected with either a lentiviral construct containing pHIV-Zsgreen+mutant *APP*, pHIV-Zsgreen+wild type *APP*, or pHIV-Zsgreen alone and allowed to grow for a week. Thereafter, cells were again dissociated and seeded at 5000 cells/well in a 24-well plate in triplicate. Cells were fed every day with ×20 media containing the appropriate amount of LDN-19389 (Selleckchem S2618). Colonies were counted after 7 days. Experiment was performed times times (1 biological replicate = 1 round of infection with subsequent experiment) in triplicate (1 technical replicate = 1 well).

## Acknowledgements

The authors thank several individuals, including Siddhartha Mitra, James Lennon, Sam Cheshier, Jordan Roselli, Stephen Ahn, Grace Hagiwara, Mike Alvarez, Pieter Both, Jami Wang, Ben W Dulken, Vincent M Alford, and Aisling Chaney. Flow cytometry analysis for this project was done on instruments in the Stanford Shared FACS Facility; BD FACSAriaII was purchased by NIH S10 shared instrumentation grant 1S10RR02933801. The authors thank the Stanford Neuroscience Microscopy Service, supported by

NIH NS069375. Funding: California Institute of Regenerative Medicine. Chan Zuckerberg Biohub. NIH R01AG059712. Harriet and CC. Tung Foundation – FR. AIRC and Marie Curie Action – BNdR – COFUND.

## Additional information

### Competing interests

Felicia Reinitz, Elizabeth Y Chen, Jane Antony, Robert C Jones, Michael F Clarke: filer of a provisional patent: U.S.Provisional Application No. 63/124,644 titled "Modulating BMP signaling in the treatment of Alzheimer's disease". Benedetta Nicolis di Robilant, Maddalena Adorno: is the co-founder of Dorian Therapeutics. Dorian therapeutics was incorporated in June 2018 and it is an early stage anti-aging company that focuses on the process of cellular senescence. Most of the experiments were performed before the company was formed. The other authors declare that no competing interests exist.

### Funding

| Funder | Grant reference number | Author |
| --- | --- | --- |
| California Institute of Regenerative Medicine | Graduate Student Fellowship | Elizabeth Y Chen |
| Chan Zucherberg Foundationg Biohub Initiative | | Elizabeth Y Chen<br>Robert C Jones<br>Sai Saroja Kolluru<br>Stephen R Quake |
| National Institutes of Health | 1R01AG059712-01 | Felicia Reinitz<br>Elizabeth Y Chen<br>Benedetta Nicolis di Robilant<br>Jane Antony<br>Neha Gubbi<br>Dalong Qian<br>Michael F Clarke |
| National Institutes of Health | AG059712 | Michael F Clarke |
| Tung Foundation | | Felicia Reinitz |
| AIRC and Marie Curie Action | | Benedetta Nicolis di Robilant |

The funders had no role in study design, data collection, and interpretation, or the decision to submit the work for publication.

### Author contributions

Felicia Reinitz, Conceptualization, Formal analysis, Investigation, Methodology, Project administration, Validation, Visualization, Writing – original draft, Writing – review and editing; Elizabeth Y Chen, Conceptualization, Data curation, Formal analysis, Investigation, Methodology, Project administration, Software, Validation, Visualization, Writing – original draft, Writing – review and editing; Benedetta Nicolis di Robilant, Conceptualization, Formal analysis, Investigation, Project administration, Validation, Visualization, Writing – original draft, Writing – review and editing; Bayarsaikhan Chuluun, Investigation, Project administration, Validation; Jane Antony, Formal analysis, Investigation, Validation, Writing – review and editing; Robert C Jones, Data curation, Project administration, Resources, Software; Neha Gubbi, Investigation, Validation; Karen Lee, Investigation; William Hai Dang Ho, Investigation, Resources; Sai Saroja Kolluru, Project administration, Resources; Dalong Qian, Funding acquisition, Project administration, Resources; Maddalena Adorno, Supervision; Katja Piltti, Aileen Anderson, Methodology, Resources; Michelle Monje, Conceptualization, Resources, Supervision, Writing – review and editing; H Craig Heller, Resources, Supervision; Stephen R Quake, Funding acquisition, Resources, Supervision; Michael F Clarke, Conceptualization, Funding acquisition, Resources, Supervision, Writing – review and editing

## Author ORCIDs
Felicia Reinitz http://orcid.org/0000-0003-4952-4457
Robert C Jones http://orcid.org/0000-0001-7235-9854
Aileen Anderson http://orcid.org/0000-0002-8203-8891
Michelle Monje http://orcid.org/0000-0002-3547-237X
H Craig Heller http://orcid.org/0000-0003-4479-5880
Stephen R Quake http://orcid.org/0000-0002-1613-0809
Michael F Clarke http://orcid.org/0000-0001-6889-4926

## Ethics
Mice were housed in accordance with the guidelines of Institutional Animal Care Use Committee. All animal procedures and behavioral studies involved in this manuscript are compliant to Stanford Administrative Panel on Laboratory Animal Care (APLAC) Protocol 10868 pre-approved by the Stanford Institutional Animal Care and Use Committee (IACUC).

## Decision letter and Author response
Decision letter https://doi.org/10.7554/eLife.66037.sa1
Author response https://doi.org/10.7554/eLife.66037.sa2

# Additional files

## Supplementary files
• Transparent reporting form

## Data availability
Datasets generated are available on Dryad Digital Repository (https://doi.org/10.5061/dryad.mpg4f4qz0 and https://doi.org/10.5061/dryad.vx0k6djtf).

The following datasets were generated:

| Author(s) | Year | Dataset title | Dataset URL | Database and Identifier |
|---|---|---|---|---|
| Chen E, Jones R, Clarke M, Quake S | 2022 | Single-Cell RNA-sequencing of neural precursor cells from an Alzheimer's mouse model, wild-type mice, and Alzheimer's mice rescued with Usp16 haploinsufficiency | https://doi.org/10.5061/dryad.mpg4f4qz0 | Dryad Digital Repository, 10.5061/dryad.mpg4f4qz0 |
| Reinitz F, Clarke M, Nicolis di Robilant B | 2022 | Microarray analysis of subventricular zone, hippocampus, and cortex from an Alzheimer's mouse model, wild-type mice, and Alzheimer's mice rescued with Usp16 haploinsufficiency | https://doi.org/10.5061/dryad.vx0k6djtf | Dryad Digital Repository, 10.5061/dryad.vx0k6djtf |

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
