## [Editor Report]

This work by Reinitz et al. provides nice evidence of a neural stem cell (NSC) defect that precedes and appears to be independent of amyloid pathology and neuroinflammation in an AD model. The authors show that targeting USP16, a BMI antagonist, can rescue some of these NSC deficits. Furthermore, scRNA-seq and GSEA points to the BMP pathway as a major player in regulating these phenotypes; in support of their hypothesis, the authors show that inhibition of BMP signaling using small molecules also rescues NSC defects. These are interesting and novel findings that will be of interest to the field.

---

## [Decision Letter]

**Decision letter after peer review:**

Thank you for submitting your article “Inhibiting USP16 rescues stem cell aging and memory in an Alzheimer's model” for consideration by *eLife*. Your article has been reviewed by three peer reviewers, including Jessica Young as the Reviewing Editor and Reviewer #1, and the evaluation has been overseen by Matt Kaeberlein as the Senior Editor. The following individual involved in review of your submission has agreed to reveal their identity: Zachary McEachin (Reviewer #2).

In general, the reviewers acknowledged the impact and novelty of the work, however concerns arose during the review process that should be addressed and/or sufficiently rebutted prior to publication in *eLife*. In particular, one reviewer brought up several point where the data seem over-interpreted and the descriptions over-simplifed. I have therefore combined the reviewer comments into two sets of revisions, one set to be addressed experimentally and one set to be addressed in writing, both as rebuttals to the reviewers and as clarifications and descriptions in the manuscript text.

Essential revisions:

Essential major points to be addressed experimentally prior to publication in *eLife*:

1. Please show the NIC/neurosphere experiments the same way in each figure. In some figures there is a table of confidence intervals, in others a graph of log fraction nonresponding and in others representative images (Figure 3E). Or please explain why the representation is different clearly in the figure legend.

2. In Figure 2: Using the please show whether this is also changed in wild type (WT) cell with age. Please show BMI expression in the APP transgenic brain and Cdkn2a expression in WT and APP mut neurospheres. Please show GFAP expression in young APP transgenic mice. It seems very surprising that already one passage abolishes Bmi1 expression (Figure 2c), and it is unclear to me how that can be interpreted as a phenotype that “accompanies aging.” Please explain.

3. In Figure 6: Please show images of GFAP staining that are represented by the quantification

4. The authors’ conclusion that Tg-SwDI/*Usp16*^+/-^ rescues cognitive defects would be strengthened by additional cognition test (three chambered assay or Barnes or Morris water maze assays).

5. The authors report an initially increased proliferation of NPCs and downstream inflammatory consequences that they have measured by GFAP reactivity and astrogliosis. However, the authors have not addressed if the increased abundance of GFAP-reactive astrocytes is instead a general lineage increase in astrocytes derived from APP-mutant/over-proliferating NPCs. Could the astrocyte reactivity rate be the same in healthy animals and it is only because there are more cells generally in the fAD mouse that the authors detect this astrogliosis signature? To address this, some attempt to quantify lineage abundances by classic markers of astrocytes, neurons, and other cells differentiated from NPCs should be included, and another measure of astrocyte reactivity, like the recently reported CD49f, to supplement GFAP.

6. The study makes a very important observation in that they show that cognitive defects in an AD mouse model can be rescued independent of plaque load. However, *Usp16* reduction (Tg-SwDI/*Usp16*^+/-^) also did not rescue astrocyte over-abundance (which the authors interpret as neuroinflammation). I thus feel that additional immunohistochemical and behavioral experiments are needed to dissect late disease-related phenotypes in Tg-SwDI/*Usp16*^+/-^ mice, and ideally also in LDN-treated mice, to be conclusive.

7. The authors elegantly use ELDA to test the “regenerative potential” of NSCs, and they suggest that an initial hyperproliferation is followed by exhaustion. However, some critical data seems to be missing to support this claim. Is neutrospheres formation increased at 1 month for example?

8. Did the authors check for any other signs of early hyperproliferation of NPCs such as SVZ thickness, general macrocephaly, etc.? Also, it remains to be determined by the authors at what stage/age hyperproliferation stops and stem cell exhaustion sets in.

9. When neutrospheres are transduced (not infected – this term is reserved for wildtype viruses), typically only the outside rim of cells become transgenic, and thus technical variation due to different sphere sizes could be a big issue. The assay (Figure 1E) should be better characterized regarding efficiency and variations to make it more convincing.

10. The authors suggest that extensive early proliferation decreases the stem cell pool in the SVZ, but they do not show data regarding the stem cell pool in 1-year-old Tg-SwDI mice. I feel an assessment of the SVZ stained for EdU, GFAP and *SOX2* should be performed at the 1-year time point, and preferentially also in the dentate gyrus to improve translatability at least to one other neural stem cell type.

11. The authors show pieces of data that lead to the conclusion that similarly reduced neurosphere numbers (Figure 2A/p6) indicate that an aging phenotype is “emulated.” Given the lack of further data comparing young and old (wt and Tg) in a rigorous manner, this statement seems overstated. In Figure 2E, it would help to interpret the age dependent nature of astrogliosis in the TgSwDI by including young animals in addition to old. Also, can the authors report neurosphere data in old mice as presented in Figure 3A/B/F?

12. In Figure 2B, it seems unclear why the authors have chosen the cortex only for this experiment. Another assay, such as a simple western blot or staining for markers for inflammation/microglial activation, would be helpful to support the interpretation.

13. The results of no increased inflammation in Figure 2F are surprising given the increased gliosis, and could be attributed to probes not included in the Luminex panel or insufficient sensitivity in the provided tissue. To address the latter, a positive control of known inflamed tissue should be included.

14. Although p16 knockdown can cause tumor formations in vivo, the authors do not have such limitations on in vitro experiments. As such, they should verify Cdkn2a’s direct involvement by specific knockdown with shRNAs and then perform neurosphere analysis.

15. The scRNAseq data are presented in a very lean manner and need more analysis. Can the authors show their scRNA data in a UMAP or tSNE plot, with cells labeled by aggregate expression of the pathways they have highlighted in Table 4? At a minimum, p-values and a volcano plot (or an appropriate alternative) of key genes should be included. Why are only gene sets presented and never individual genes? There could be some interesting changes looking at differential expression between single genes instead of aggregates of genes. With this data, the authors have an opportunity to examine the relationship between Cdkn2a and USP16 expression, do they see these more co-expression in the Tg mouse? Is there transcriptional evidence of the hyperproliferation? This dataset would also be useful to query for inflammatory markers in the NSCs.

16. In regard to the RNA-seq data: Can the authors please comment if there are other genes regulating cellular senescence that are differentially expressed? Is aberrant Cdkn2a expression replicated here?

Essential major points to be addressed in writing in the manuscript prior to publication in *eLife*:

1. In the discussion: please comment on how this might relate to late-onset, sporadic AD which is not caused by mutations in APP. Especially as decreased neurogenesis is seen in SAD brains (PMID: 30911133)

2. Given that Bmi-1 expression is reduced in the Tg-SwDI mice (Figure 2D) and loss of Bmi-1 results in dramatic reduction of neurosphere size (Molofsky et al., 2005, Nature), is there a difference in the size (or shape) of neurospheres from wild type (WT) and Tg-SwDI mice?

3. Is an increase in NIC capacity a general result of Cdkna2a loss? Do WT/Cdkn2a-/- show similar or increased NIC capacity as WT at 3 months (Figure 3A)?

4. In the Tg-SwDI/*Usp16*^+/-^ is Bmi-1 expression changed (perhaps as a compensatory mechanism of *Usp16* loss). Could this explain the only partial rescue of NIC frequency (Figure 3F).

5. In Figure 5E, is this quantification from the ELDA assay or a different assay? If different, ELDA should be used to remain consistent.

6. While the authors show quantitative data of Thioflavin staining in Figure 6B, representative images should be included in figure or supplementary. Similarly, representative images of GFAP staining for Figure 6A should be included.

7. The authors should be more overall more critical with their own data, and explain what aspects of their model system are likely useful and can be translated to a human situation, and which ones are not. I would be very hesitant to commit to what extent the mouse SVZ relates to the human situation. A phenotype detected in a three to four-month-old transgenic mice might be translatable to early human development (fetal/early life), to adult neurogenesis in early AD pathology (late in life), or not directly translatable. Mouse and human NPCs are extremely different cells (see work from Conti L. et al., Sun et al., Elkabetz et al., etc…), and this is also eident from the data here (e.g. compare scales in Figure 1B and 1E). I suggest the authors to please avoid rushed over-interpretation.

8. It is not clear how the authors come to the conclusion that this “suggests that the self-renewal defect […] not specific to the Swedish, Dutch and Iowa mutations, but also more broadly seen with other APP mutations” (Figure 1E/p5). They did not test different expression/protein levels of WT, the individual mutations, or combinations, which would be necessary to make such a statement.

9. The attempt for assessing the effect of mutant APP on NPC phenotypes in the human system is a strength of this study. In Figure 1E/p5, the authors claim that “the effects observed in NPCs derived from a genetic mouse model can be robustly recapitulated in human NPCs expressing mutant APP.” As to the APP setting, both models are very different, in that human cells possess two copies of endogenous APP. “Fetal human NPCs” are not very well defined, an no characterization of endogenous/transgenic APP expression/protein levels are provided here. The choice of this models seems suboptimal. Also, to draw meaningful conclusions, the effect should be tested on different genetic backgrounds and well-characterized NPCs (e.g. from APP-mutant iPSCs). The conclusion that the “effects observed in NPCs derived from a genetic mouse model can be robustly recapitulated in human NPCs” is thus not appropriate and probably vastly oversimplified.

10. The authors show that the neurosphere phenotype of NPCs of 3-month-old mice that was rescued by Cdkn2a-KO. This is followed by the statement (line158) that APP mutations accelerate aging through Cdkn2a. In the context pf the presented data, this is a prominent overstatement.

*Reviewer #1 (Recommendations for the authors):*

In general, the data supports the conclusions and I think this work brings several novel aspects to the field. The comments below are for consistency in how the data is presented.

Please show the NIC/neurosphere experiments the same way in each figure. In some figures, there is a table of confidence intervals, in others a graph of log fraction nonresponding and in others representative images (Figure 3E). Or please explain why the representation is different clearly in the figure legend.

In Figure 2: I would like to also see the NPC proliferation assays, is this changed in WT cell with age. Please show BMI expression in the APP transgenic brain if possible and Cdkn2a expression in WT and APP mut neurospheres. Please show GFAP expression in young APP transgenic mice.

In Figure 6: Please show images of GFAP staining that are represented by the quantification

In regard to the RNA-seq data: Can the authors please comment if there are other genes regulating cellular senescence that are differentially expressed? Is aberrant Cdkn2a expression replicated here?

In the discussion: please comment on how this might relate to late-onset, sporadic AD which is not caused by mutations in APP. Especially as decreased neurogenesis is seen in SAD brains (PMID: 30911133)

*Reviewer #2 (Recommendations for the authors):*

1) Given that Bmi-1 expression is reduced in the Tg-SwDI mice (Figure 2D) and loss of Bmi-1 results in dramatic reduction of neurosphere size (Molofsky et al., 2005, Nature), is there a difference in the size (or shape) of neurospheres from WT and Tg-SwDI mice?

2) Is an increase in NIC capacity a general result of Cdkna2a loss? Do WT/Cdkn2a-/- show similar or increased NIC capacity as WT at 3 months (Figure 3A)?

3) In the Tg-SwDI/*Usp16*^+/-^ is Bmi-1 expression changed (perhaps as a compensatory mechanism of *Usp16* loss). Could this explain the only partial rescue of NIC frequency (Figure 3F).

4) In Figure 5E, is this quantification from the ELDA assay or a different assay? If different, ELDA should be used to remain consistent.

5) While the authors show quantitative data of Thioflavin staining in Figure 6B, representative images should be included in figure or supplementary. Similarly, representative images of GFAP staining for Figure 6A should be included.

6) The authors’ conclusion that Tg-SwDI/*Usp16*^+/-^ rescues cognitive defects would be strengthened by additional cognition test (three chambered assay or Barnes or Morris water maze assays).

*Reviewer #3 (Recommendations for the authors):*

In the present manuscript entitled “Inhibiting USP16 rescues stem cell aging and memory in an Alzheimer's model,” Clarke and colleagues describe early NPC defects seen in Tg-SwDI mouse model of familial AD. Interestingly, the described phenotypes, that comprise early stem cell exhaustion followed by impaired self-renewal, clearly pre-date later inflammatory, neurodegenerative classically features typically described as disease-related in mouse models. They next position Cdkn2a and BmiI at the mechanistic core of this early NPC phenotype, and present USP16 and BMP inhibition as potential therapeutic targets. The manuscript is overall well-written and figures are clear and understandable. A particular strength of the manuscript is the dogma-free approach and the general frame of the study is highly innovative, relevant and interesting. However, the authors make several assumptions that are not sufficiently backed-up by the data, and overall appear to be drawn to large overinterpretations, which have dampened this reviewer’s initial enthusiasm. Also, the manuscript reads in a way that even the authors themselves have not really decided what role they think Cdkn2a is playing here, and thus in the end it simply appears as a gene that can inhibit cell proliferation, of which they detect more in the Tg AD mouse model, and the reader is left with the feeling that probably anti-proliferative perturbation would have done the job. Further, the authors’ schematic hints at that Cdkn2a is involved in the astrogliosis process, but the current manuscript fails to present any solid data supporting that claim. Below I have listed several major points that render the paper inappropriate for publication in *eLife*. My points are intended to help the authors improve their manuscript for publication in *eLife*, but I currently feel that this would take some tremendous effort to address my points with new and conclusive experimental data.

1. The authors report an initially increased proliferation of NPCs and downstream inflammatory consequences that they have measured by GFAP reactivity and astrogliosis. However, the authors have not addressed if the increased abundance of GFAP-reactive astrocytes is instead a general lineage increase in astrocytes derived from APP-mutant/over-proliferating NPCs. Could the astrocyte reactivity rate be the same in healthy animals and it is only because there are more cells generally in the fAD mouse that the authors detect this astrogliosis signature? To address this, some attempt to quantify lineage abundances by classic markers of astrocytes, neurons, and other cells differentiated from NPCs should be included, and another measure of astrocyte reactivity, like the recently reported CD49f, to supplement GFAP.

2. The study makes a very important observation in that they show that cognitive defects in an AD mouse model can be rescued independent of plaque load. However, *Usp16* reduction (Tg-SwDI/*Usp16*^+/-^) also did not rescue astrocyte over-abundance (which the authors interpret as neuroinflammation). I thus feel that additional immunohistochemical and behavioral experiments are needed to dissect late disease-related phenotypes in Tg-SwDI/*Usp16*^+/-^ mice, and ideally also in LDN-treated mice, to be conclusive.

3. The authors elegantly use ELDA to test the ‘regenerative potential’ of NSCs, and they suggest that an initial hyperproliferation is followed by exhaustion. However, some critical data seems to be missing to support this claim. Is neutrospheres formation increased at 1 month for example?

4. Did the authors check for any other signs of early hyperproliferation of NPCs such as SVZ thickness, general macrocephaly, etc.? Also, it remains to be determined by the authors at what stage/age hyperproliferation stops and stem cell exhaustion sets in.

5. The authors should be more overall more critical with their own data, and explain what aspects of their model system are likely useful and can be translated to a human situation, and which ones are not. I would be very hesitant to commit to what extent the mouse SVZ relates to the human situation. A phenotype detected in a three to four-month-old transgenic mice might be translatable to early human development (fetal/early life), to adult neurogenesis in early AD pathology (late in life), or not directly translatable. Mouse and human NPCs are extremely different cells (see work from Conti L. et al., Sun et al., Elkabetz et al., etc…), and this is also eident from the data here (e.g. compare scales in Figure 1B and 1E). I suggest the authors to please avoid rushed over-interpretation.

6. It is not clear how the authors come to the conclusion that this “suggests that the self-renewal defect […] not specific to the Swedish, Dutch and Iowa mutations, but also more broadly seen with other APP mutations” (Figure 1E/p5). They did not test different expression/protein levels of WT, the individual mutations, or combinations, which would be necessary to make such a statement.

7. The attempt for assessing the effect of mutant APP on NPC phenotypes in the human system is a strength of this study. In Figure 1E/p5 the authors claim that “the effects observed in NPCs derived from a genetic mouse model can be robustly recapitulated in human NPCs expressing mutant APP.” As to the APP setting, both models are very different, in that human cells possess two copies of endogenous APP. “Fetal human NPCs” are not very well defined, an no characterization of endogenous/transgenic APP expression/protein levels are provided here. The choice of this models seems suboptimal. Also, to draw meaningful conclusions, the effect should be tested on different genetic backgrounds and well-characterized NPCs (e.g. from APP-mutant iPSCs). The conclusion that the “effects observed in NPCs derived from a genetic mouse model can be robustly recapitulated in human NPCs” is thus not appropriate and probably vastly oversimplified.

8. When neutrospheres are transduced (not infected – this term is reserved for wildtype viruses), typically only the outside rim of cells become transgenic, and thus technical variation due to different sphere sizes could be a big issue. The assay (Figure 1E) should be better characterized regarding efficiency and variations to make it more convincing.

9. The authors suggest that extensive early proliferation decreases the stem cell pool in the SVZ, but they don't show data regarding the stem cell pool in 1-year-old Tg-SwDI mice. I feel an assessment of the SVZ stained for EdU, GFAP, and *SOX2* should be performed at the 1-year time point, and preferentially also in the dentate gyrus to improve translatability at least to one other neural stem cell type.

10. The authors show pieces of data that lead to the conclusion that similarly reduced neurosphere numbers (Figure 2A/p6) indicate that an aging phenotype is “emulated.” Given the lack of further data comparing young and old (WT and Tg) in a rigorous manner, this statement seems overstated. In Figure 2E, it would help to interpret the age dependent nature of astrogliosis in the TgSwDI by including young animals in addition to old. Also, can the authors report neurosphere data in old mice as presented in Figure 3A/B/F?

11. In Figure 2B, it seems unclear why the authors have chosen the cortex only for this experiment. Another assay, such as a simple western blot or staining for markers for inflammation/microglial activation, would be helpful to support the interpretation.

12. It seems very surprising that already one passage abolishes Bmi1 expression (Figure 2c), and it is unclear to me how that can be interpreted as a phenotype that “accompanies aging".” Please explain.

13. The results of no increased inflammation in Figure 2F are surprising given the increased gliosis, and could be attributed to probes not included in the Luminex panel or insufficient sensitivity in the provided tissue. To address the latter, a positive control of known inflamed tissue should be included.

14. The authors show that the neurosphere phenotype of NPCs of 3-month-old mice that was rescued by Cdkn2a-KO. This is followed by the statement (line158) that APP mutations accelerate aging through Cdkn2a. In the context pf the presented data, this is a prominent overstatement.

15. Although p16 knockdown can cause tumor formations in vivo, the authors do not have such limitations on in vitro experiments. As such, they should verify Cdkn2a’s direct involvement by specific knockdown with shRNAs and then perform neurosphere analysis.

16. The scRNAseq data are presented in a very lean manner and need more analysis. Can the authors show their scRNA data in a UMAP or tSNE plot, with cells labeled by aggregate expression of the pathways they have highlighted in Table 4? At a minimum, p-values and a volcano plot (or an appropriate alternative) of key genes should be included. Why are only gene sets presented and never individual genes? There could be some interesting changes looking at differential expression between single genes instead of aggregates of genes. With this data, the authors have an opportunity to examine the relationship between Cdkn2a and USP16 expression, do they see these more co-expression in the Tg mouse? Is there transcriptional evidence of the hyperproliferation? This dataset would also be useful to query for inflammatory markers in the NSCs.

---

## [Author Response]

Essential revisions:Essential Major points to be addressed experimentally prior to publication in eLife:1. Please show the NIC/neurosphere experiments the same way in each figure. In some figures there is a table of confidence intervals, in others a graph of log fraction nonresponding and in others representative images (Figure 3E). Or please explain why the representation is different clearly in the figure legend.

This has been corrected to bar graphs with corresponding tables of confidence intervals.

2. In Figure 2: Using the please show whether this is also changed in WT cell with age. Please show BMI expression in the APP transgenic brain and Cdkn2a expression in WT and APP mut neurospheres. Please show GFAP expression in young APP transgenic mice. It seems very surprising that already one passage abolishes Bmi1 expression (Figure 2c), and it is unclear to me how that can be interpreted as a phenotype that "accompanies aging". Please explain.

When directly comparing the young vs aged WT NIC frequencies, we do not see a significant difference in self-renewal, however, there is a significant decrease in self-renewal capacity with aging in the Tg-SwDI neurospheres. We have not included this data in the updated manuscript, but have included the analysis in Author response image 1.

**Author response image 1. sa2fig1:** 

As requested, we have measured *Bmi1* expression in the Tg-SwDI transgenic brain as shown in Author response image 2; we did not see any difference in the expression of *Bmi1* in the cerebral cortex. We have added *Cdkn2a* expression in WT and Tg-SwDI neurospheres to Figure 1C in our revised paper.

We also stained brain sections from 3-4 month old WT and Tg-SwDI mice for GFAP expression. Throughout the cerebral cortex there were sparse regions of aggregated GFAP+ cells, that we called “GFAP clusters”. These GFAP clusters comprise up to 25-30 GFAP+ cells. As a possible indication of early inflammation, we counted the number of clusters, but we observed no differences between the genotypes in this age group. This data has been included in the revised manuscript (Figure 2 —figure supplement 5).We would like to clarify that when we had described a phenotype that “accompanies aging” in the text (previously line 133), we were referring to the decreased capacity of SVZ derived NPCs to form neurospheres in older mice compared to younger mice. We have updated this section of the text to use more clear descriptions that more accurately reflect the data.

Additionally, in reference to the reviewer’s concern about *Bmi1* expression, others have shown an increased expression of *Cdkn2a* and decreased expression of *Bmi1* in aging brain as well, and the citations to these references have now been added to this section of the text in the updated manuscript.

3. In Figure 6: Please show images of GFAP staining that are represented by the quantification

These have been added to Supplementary Figure 6 —figure supplement 1.

4. The authors' conclusion that Tg-SwDI/*Usp16*^+/-^ rescues cognitive defects would be strengthened by additional cognition test (three chambered assay or Barnes or Morris water maze assays).

As an addition to our novel object recognition (NOR) test, we performed Barnes maze testing, and these data can be found in Supplementary Figure 6 —figure supplement 3. Consistent with the NOR experiment, this data supports long term and spatial memory deficits are present in the Tg-SwDI mice which are also rescued by *Usp16* haploinsufficiency.

5. The authors report an initially increased proliferation of NPCs and downstream inflammatory consequences that they have measured by GFAP reactivity and astrogliosis. However, the authors have not addressed if the increased abundance of GFAP-reactive astrocytes is instead a general lineage increase in astrocytes derived from APP-mutant/over-proliferating NPCs. Could the astrocyte reactivity rate be the same in healthy animals and it is only because there are more cells generally in the fAD mouse that the authors detect this astrogliosis signature? To address this, some attempt to quantify lineage abundances by classic markers of astrocytes, neurons, and other cells differentiated from NPCs should be included, and another measure of astrocyte reactivity, like the recently reported CD49f, to supplement GFAP.

The reviewer brings up an important distinction between increased GFAP-reactive astrocytes versus an increase in number of cells generally in the Tg-SwDI mouse. To further investigate reactive astrogliosis and to quantify numbers of astrocytes we performed a number of studies, including qPCR of known markers of reactive astrocytes (A1 astrocytes) vs non-reactive A2 astrocytes (Zamanian et al., 2012), microarray, and differentiation studies. In the cortex of one year old Tg-SwDI mice, there was not a significant increase in A1 astrocyte markers (Figure 2 —figure supplement 2). However, using a microarray of 2 year old mice, and analyzing the same A1 and A2 astrocyte markers, we did see an increase in the A1 astrocytic markers as seen in Figure 2 —figure supplement 4, suggesting there is eventually reactive astrogliosis but further aging is required. Additionally, we placed neurospheres derived from the SVZ of Tg-SwDI mice and WT controls into differentiation conditions and analyzed the number of GFAP positive cells following differentiation (included in Figure 2 —figure supplement 3). We found that cells derived from Tg-SwDI mice formed more GFAP-positive cells than those derived from wild-type controls, suggesting at least an increase in production of astrocytes when controlling for total number of cells. All these data taken together suggest that in mutant Tg-SwDI mice, there are more astrocytes produced and that these astrocytes become reactive with aging.

The reviewer points out using CD49f as another measure of astrocyte reactivity to supplement GFAP. This marker was recently reported by Barber et al. as a “…reactivity-independent marker (expressed in both unstimulated and reactive astrocytes)…”. (Barbar et al., 2020) Thus, CD49f is a novel marker shown to be useful in identification and isolation of human and iPSC-derived astrocytes that can be stimulated to become A1-like reactive astrocytes. However, this marker cannot differentiate between reactive vs non-reactive astrocytes. Furthermore, the authors report that when they tested CD49f on “whole mouse brain of *ALDH1L1*^eGFP^, no CD49f^+^ astrocytes were ALDH1L1^+^, suggesting that CD49f could be human specific.”

To look for astrocyte reactivity, we instead analyzed mRNA using the verified A1 specific astrocyte markers as described above and differentiation followed by astrocyte quantification immunohistochemically, as shown in Figure 2—figure supplement 4.

6. The study makes a very important observation in that they show that cognitive defects in an AD mouse model can be rescued independent of plaque load. However, *Usp16* reduction (Tg-SwDI/*Usp16*^+/-^) also did not rescue astrocyte over-abundance (which the authors interpret as neuroinflammation). I thus feel that additional immunohistochemical and behavioral experiments are needed to dissect late disease-related phenotypes in Tg-SwDI/*Usp16*^+/-^ mice, and ideally also in LDN-treated mice, to be conclusive.

To dissect late disease-related phenotypes in the Tg-SwDI/*Usp16^+/-^* mice, we conducted a variety of assays, including immunohistochemical quantification of GFAP+ cells in the cerebral cortex, thioflavin S staining for amyloid plaques, behavioral cognitive testing with novel object recognition and Barnes Maze tests, and a Luminex assay to study inflammation. In Figure 6 we show that *Usp16* reduction also decreases the number of GFAP+ cells (Figure 6A, Figure 6 —figure supplement 1) as well as cognitive defects (Figure 6C) even though there is no change in amyloid plaque burden (Figure 6B) or inflammatory markers (Figure 6 —figure supplement 2).

Although testing LDN-treated mice would be ideal, there are several factors that make this difficult as it has never been studied in the context of an intracranial application, thus a study would require an analysis of blood-brain barrier penetration, dose optimization, and determining the appropriate length/frequency of treatment. We therefore opted to use an in vitro study instead to better understand involvement of the BMP pathway, which also afforded us the advantage of using human cells.

Finally, while we agree with the reviewer that it would be ideal to continue to dissect late disease-related phenotypes in Tg-SwDI/*Usp16^+/-^* mice, the benefit of our conclusions come from identifying an earlier disease-related phenotype potentially related to cognitive decline that may be intervenable prior to the onset of late disease-related phenotypes.

7. The authors elegantly use ELDA to test the 'regenerative potential' of NSCs, and they suggest that an initial hyperproliferation is followed by exhaustion. However, some critical data seems to be missing to support this claim. Is neutrospheres formation increased at 1 month for example?

As recommended by the reviewers, we assessed neurosphere formation at 1 month old and found that there was no significant difference in the stem cell frequencies between WT and Tg-SwDI. This suggests that self-renewal changes are not detectable early in life and require some degree of aging before this phenotype can be elicited.

**Author response image 3. sa2fig3:** 

8. Did the authors check for any other signs of early hyperproliferation of NPCs such as SVZ thickness, general macrocephaly, etc.? Also, it remains to be determined by the authors at what stage/age hyperproliferation stops and stem cell exhaustion sets in.

Given the additional ELDA experiments completed in response to number 7 above and the immunohistochemical staining of older mice (see response to number 10 below) completed since the time of this manuscript’s submission, it appears as though hyperproliferation and stem cell exhaustion begin at around 3 months old and end at about 1 year old. As shown in the response to number 7, we do not see differences in stem cell frequencies at 1 month old, but as the manuscript shows, this difference is evident at 3 months old (Figure 1B) alongside increased proliferation (Figure 1A). We no longer see differences in proliferation at 1 year old (Figure 2A), yet we continue to see decreased stem cell frequencies in cells derived from the Tg-SwDI subventricular zone (Figure 2B). Taken together, hyperproliferation extends through stem cell exhaustion, ending by approximately 1 year of age. This is not unexpected as we know that hyperproliferation and stem cell exhaustion are dynamic processes that occur over time, thus there is not necessarily a time at which one would see that hyperproliferation abruptly ends and stem cell exhaustion begins, as they are occurring simultaneously (Molofsky et al., 2003). We have updated Figure 6D to reflect this timeline.

Moreover, we performed MRI and MRS (Magnetic resonance spectroscopy), typically used to diagnose AD in patients, on our mouse models. In particular, proton magnetic resonance spectroscopy (MRS) provides a window into the biochemical changes associated with the loss of neuronal integrity that involve the brain before the manifestations of cognitive impairment in patients who are at risk for Alzheimer’s disease. Given that our Alzheimer’s mouse model, at least anatomically, does not present the same macroscopic neurodegeneration observed in patients, we sought to take advantage of MRS techniques to investigate the biochemical changes and neurodegeneration that usually appear before anatomical manifestations. We analyzed 18-22 month old mice by MRS and normalized each metabolite’s expression by the “sum of metabolites” (data were comparable when normalization was done by Creatinine). As it is possible to observe in Author response image 4, we did not observe any metabolic differences between the genotypes. NAA and myo-inositol (mIns), are the most used metabolites for detecting dementia in humans. (abbreviations: SOM: Sum of metabolites; NAA: N-acetylaspartate; mIns: Myo-Inositol; Tau: Taurine; Cho: choline; PC: phosphocholine; GLU: glutamate; GLN: glutamine)

**Author response image 4. sa2fig4:** 

9. When neutrospheres are transduced (not infected – this term is reserved for wildtype viruses), typically only the outside rim of cells become transgenic, and thus technical variation due to different sphere sizes could be a big issue. The assay (Figure 1E) should be better characterized regarding efficiency and variations to make it more convincing.

The human fetal neural stem cells were first dissociated into single cells and then transduced with a high enough titer to ensure 100% transduction of cells. Therefore, they are green throughout (Figure 5D).

10. The authors suggest that extensive early proliferation decreases the stem cell pool in the SVZ, but they don't show data regarding the stem cell pool in 1-year-old Tg-SwDI mice. I feel an assessment of the SVZ stained for EdU, GFAP and SOX2 should be performed at the 1-year time point, and preferentially also in the dentate gyrus to improve translatability at least to one other neural stem cell type.

We thank the reviewer for making this astute observation. Since the submission of this manuscript, we have assessed the number of EdU/GFAP/*SOX2* positive cells in the SVZ of 1-year old mice (see Author response image 5) and have found no significant difference between Tg-SwDI and WT mice. As discussed above in response to query number 8, the decrease in number of proliferating NPCs (or lack of hyperproliferation) at 1 year old correlates with decreased self-renewal over time beginning at 3 months old and continuing through 1 year old (Figure 1B and 2B, respectively).

**Author response image 5. sa2fig5:** 

Total number of EdU/*SOX2*/GFAP triple positive cells in 1 year old mice (N = 3 WT and N = 4 Tg-SwDI mice)Shown in Author response image 6 is a representative image of the subgranular zone (SGZ) of the hippocampal dentate gyrus of 3-4 month old mice (*SOX2* – red, GFAP – cyan, EdU – green, Dapi – blue). The EdU+*Sox2Sox2*^+^GFAP+ triple positive cells were counted in the subgranular zone (SGZ) using stereology shown in the graph on the right, n = 3. Unlike the SVZ, we did not observe a significant difference in proliferating NPCs marked by EdU, *SOX2* and GFAP. This may be due to differences in the expression of markers that characterize NPCs that are found in the hippocampus compared to the SVZ. It is therefore challenging to directly compare NPCs of the SVZ and the SGZ of the hippocampal dentate gyrus, and we have chosen not to include this data in the manuscript.

**Author response image 6. sa2fig6:** 

11. The authors show pieces of data that lead to the conclusion that similarly reduced neurosphere numbers (Figure 2A/p6) indicate that an aging phenotype is "emulated". Given the lack of further data comparing young and old (wt and Tg) in a rigorous manner, this statement seems overstated. In Figure 2E it would help to interpret the age dependent nature of astrogliosis in the TgSwDI by including young animals in addition to old. Also, can the authors report neurosphere data in old mice as presented in Figure 3A/B/F?

Please refer to our response to reviewer comment 2 above, which addresses the question of our data emulating an aging phenotype; there you will also find our measurement of GFAP clusters in 3-4 month old TgSwDI mice compared to WT which has been included in our revised paper as Figure 2 —figure supplement 5.

When directly comparing data from the young vs aged wild-type (WT) NIC frequencies, we did not see a significant difference in self-renewal, however, there is a significant decrease in self-renewal capacity with aging in the Tg-SwDI neurospheres (Data included in response #2 above).

Lastly, the reviewer makes an excellent suggestion to show neurosphere initiating capacity in older *Cdkn2a* knockout mice in Figure 3; unfortunately, these mice form tumors soon after age 3-4 months (by 9 months most of these mice have large tumors and must be euthanized). Thus, the limiting dilution experiments cannot be performed in aged mice due to the number of mice needed to survive to this age to carry out the experiment.

12. In Figure 2B. it seems unclear why the authors have chosen the cortex only for this experiment. Another assay, such as a simple western blot or staining for markers for inflammation/microglial activation, would be helpful to support the interpretation.

*Cdkn2a* expression can be difficult to detect in a heterogeneous population: its expression is both transient and limited to few cells, many of which are destined to die due to apoptosis and be rapidly eliminated. We tested *Cdkn2a* expression in three different regions of the brain in 2 year old mice as shown in Author response image 7 (n=6): SVZ, DG, and Cortex (p value between WT and Tg-SwDI DG is 0.05; p value between WT and Tg-SwDI Cortex is 0.015). Even though Tg-SwDI mice showed an increased expression in all the regions analyzed, *Cdkn2a* expression levels for WT mice in SVZ and DG were very low (<0.0001), making it nearly impossible to confidently detect differences in mRNA expression. As a consequence of this sensitivity issue in the SVZ and DG, we decided to present data from the cortex only, as the expression is higher overall. Our results corroborate studies done by Molofsky et al., for instance, where they were not able to detect *Cdkn2a* expression by PCR in 60-day-old mice and similarly, could not detect p16INK4a protein in the SVZ by western blot, consistent with other studies (Krishnamurthy et al., 2004; Molofsky et al., 2006; Zindy, Quelle, Roussel, and Sherr, 1997) due to its transient low expression level and limited sensitivity of available antibodies.

**Author response image 7. sa2fig7:** 

13. The results of no increased inflammation in Figure 2F are surprising given the increased gliosis, and could be attributed to probes not included in the Luminex panel or insufficient sensitivity in the provided tissue. To address the latter, a positive control of known inflamed tissue should be included.

We agree with the reviewer that the result in Figure 2F was initially a surprise to us as well. To ensure that the results we were seeing were real and not due to insufficient sensitivity or insufficient probes, we tested levels of inflammation in 2 year old Tg-SwDI mice (n=3) and found that there were indeed significantly higher levels of only certain cytokines expressed over the 1 year old mice we showed in our paper. This is shown as an added positive control and has been included in Supplementary Figure 6 —figure supplement 2.

14. Although p16 knockdown can cause tumor formations in vivo, the authors do not have such limitations on in vitro experiments. As such, they should verify Cdkn2a's direct involvement by specific knockdown with shRNAs and then perform neurosphere analysis.

We did not complete the recommended experiment for several reasons: (1) by genetically knocking out *Cdkn2a* expression in Tg-SwDI mice and performing our neurosphere limiting dilution analysis, we have demonstrated *Cdkn2a*’s direct involvement in neurosphere formation in a manner that would be more effective than the use of shRNAs against *Cdkn2a*. To strengthen our argument, we point to similar studies (Bruggeman et al., 2005; Molofsky, He, Bydon, Morrison, and Pardal, 2005; Molofsky et al., 2006) that have demonstrated a significant increase in self-renewal of neural stem cells in both WT and *Bmi1-/-* mice by also genetically knocking out *Cdkn2a*. (2) Considering that *Cdkn2a* is already very lowly expressed in neurospheres, it would be difficult to verify that the knockdown occurred.

15. The scRNAseq data are presented in a very lean manner and need more analysis. Can the authors show their scRNA data in a UMAP or tSNE plot, with cells labeled by aggregate expression of the pathways they have highlighted in Table 4? At a minimum, p-values and a volcano plot (or an appropriate alternative) of key genes should be included. Why are only gene sets presented and never individual genes? There could be some interesting changes looking at differential expression between single genes instead of aggregates of genes. With this data, the authors have an opportunity to examine the relationship between Cdkn2a and USP16 expression, do they see these more co-expression in the Tg mouse? Is there transcriptional evidence of the hyperproliferation? This dataset would also be useful to query for inflammatory markers in the NSCs.

We have conducted a more thorough analysis of the data and present two additional supplementary figures addressing the reviewer’s points that show the appropriate tSNE (Figure 4 —figure supplement 1) and volcano plots (Figure 4 —figure supplement 4) with additional panels of hyperproliferation (Figure 4 —figure supplement 3) and inflammatory markers (Figure 4 —figure supplement 2). The volcano plot actually bolsters our original argument that looking at gene sets is far more ideal than looking at individual genes. Oftentimes, when we are looking at individual differentially expressed genes, it can be difficult to choose which ones to pursue in a therapeutic context. The nature of GSEA is to highlight groups of genes that are significantly enriched and related in the same pathway (which can be more easily targeted), rather than to find individual genes that are significantly enriched. Furthermore, it may be difficult to see whether individual genes that are upregulated or downregulated are due to a compensatory effect rather than contributing to the pathophysiology of the disease. With regards to *Cdkn2a*, neither the 3-4 month old or 1 year old WT and Tg-SwDI expressed this gene in at least 1% of cells at a minimum logFC of 0.01. While there were no significant differences in expression between 3-4 month old and 1 year old Tg-SwDI and WT mice for either *Bmi1* or *Usp16* (see Author response image 8) using a Wilcoxon rank Sum test for differential gene expression, we did see a mild upregulation in expression of both genes in Tg-SwDI mice compared to WT with greater expression of *Usp16* over *Bmi1* at both ages. When we surveyed inflammatory markers, similar to what our Luminex assay showed, we did not see a substantial increase in any of the markers at both the 3-4 month and 1 year old time point (only *Csf1* and *Il18* were upregulated at the 3-4 month old time point in Tg-SwDI and only *Lifr* and *Ifngr1* were upregulated at the 1 year old time point) (Figures 4 —figure supplement 2). A possible explanation for this result could be that we were selecting for a cell population (within the SVZ) that itself does not exhibit inflammation at either time point. Finally, while we were able to survey several proliferation and cell-cycle related genes, only a few were differentially expressed. We include these in the same supplementary figures (Figure 4 —figure supplement 3). Notably, *Anapc2* was significantly upregulated while cell cycle inhibitor *Cdkn1a* was significantly downregulated in 3-4 month old Tg-SwDI compared to WT. At 1 year of age, only *Cdkn1b* was significantly downregulated in 1 year old Tg-SwDI mice compared to WT.

**Author response image 8. sa2fig8:** 

16. In regards to the RNA-seq data: Can the authors please comment if there are other genes regulating cellular senescence that are differentially expressed? Is aberrant Cdkn2a expression replicated here?

*Cdkn2a* expression was too lowly expressed and not differentially expressed in our single-cell RNA-seq data. In looking at a panel of genes commonly known to regulate senescence (*Trp53, Cdkn1a, Mtor, Crebbp, Ep300, Mdm2, Btg2, E2f1, E2f2, E2f3, E2f4*, and *E2f7*) (Kumari and Jat, 2021), genes that were differentially expressed in 3-4 month old mice (Tg-SwDI vs WT) included:

**Author response table 1. sa2table1:** 

Gene	Avg_logFC difference	% of Tg-SwDI cells	% of WT cells	P_val_adj
*Trp53*	0.7315253	0.155	0.107	0
*Cdkn1a (p21)*	-0.2627055	0.057	0.170	0
*Mtor*	0.8106869	0.169	0.161	0
*E2f4*	0.3536161	0.125	0.085	3.987E-132

And in 1 year old mice:

**Author response table 2. sa2table2:** 

Gene	Avg_logFC difference	**% of Tg-SwDI cells**	% of WT cells	P_val_adj
*Crebbp*	-0.3303011	0.394	0.409	0

Interestingly, while *Trp53*, *Mtor*, and *E2f4* were increased in young Tg-SwDI mice compared to WT mice suggesting more senescence (all of which are implicated in activating senescence), *p21* expression (which is usually up in senescence) was downregulated in 3-4 month old Tg-SwDI mice. In 1 year old mice, the only senescence related gene that was differentially expressed was *Crebbp* which was downregulated in Tg-SwDI mice, suggesting an increase in senescence and decrease in proliferation.

Essential Major points to be addressed in writing in the manuscript prior to publication in eLife:1. In the discussion: please comment on how this might relate to late-onset, sporadic AD which is not caused by mutations in APP. Especially as decreased neurogenesis is seen in SAD brains (PMID: 30911133)

We have updated the discussion to address this.

To expand on this, there are some encouraging features in the Tg-SwDI mouse model including the presence of cognitive defects after the accumulation of amyloid β plaques, which occurs similarly in humans with both dominantly inherited AD and late-onset AD (LOAD) (Bateman et al., 2012; Jack et al., 2010). Bateman et al., conducted a study with 128 AD mutation carriers and found that Aß_42_ concentrations in the CSF declined 25 years before expected symptom onset (estimated as patient’s age at onset) while Aß deposition as measured by PET scans with the use of Pittsburgh compound B was detected 15 years before expected symptom onset (Bateman et al., 2012). These same carriers did not have significant differences in mini-mental state examination scores until 5 years before expected symptom onset or significant differences in the logical memory test until 10 years before expected symptom onset (Bateman et al., 2012).

In another study characterizing sporadic AD, Villemagne et al. 2012 also used Pittsburgh compound B PET studies to study those with dementia of the Alzheimer type (DAT), those with mild cognitive impairment (MCI), and age-matched healthy controls (Villemagne et al., 2011). At baseline, 31% of healthy control subjects showed high PiB retention indicating Aß deposition and of these, 16% developed MCI or DAT by 20 months and 25% by 3 years (Villemagne et al., 2011). A similar study by Morris et al. 2009 also found that in a study of 159 participants, the mean cortical binding potential for PiB assessed with PET was also a significant predictor of progression from cognitive normality to DAT (Morris et al., 2009). Furthermore, studies such as those by Salloway et al. 2014 found that treating the plaques alone did not rescue cognitive defects (Salloway et al., 2014).

All of these studies highlight the clinical similarities of late-onset sporadic AD and the importance of early therapeutic intervention, and our work reveals a new avenue of therapeutic approach that could apply to both sporadic and familial AD based on their shared features, though confirmatory studies in humans must be pursued to confirm this.

2. Given that Bmi-1 expression is reduced in the Tg-SwDI mice (Figure 2D) and loss of Bmi-1 results in dramatic reduction of neurosphere size (Molofsky et al., 2005, Nature), is there a difference in the size (or shape) of neurospheres from WT and Tg-SwDI mice?

After performing *ad hoc* analysis of neurosphere diameter in 3-4 month old mice, we did not observe a significant difference between the size of neurospheres derived from the SVZ of Tg-SwDI vs WT mice, shown in Author response image 9 (n=3 biological replicates). Nor did we observe any qualitative difference in neurosphere shape (see accompanying 4x images).

**Author response image 9. sa2fig9:** 

3. Is an increase in NIC capacity a general result of Cdkna2a loss? Do WT/Cdkn2a-/- show similar or increased NIC capacity as WT at 3 months (Figure 3A)?

NIC capacity is generally increased as a result of *Cdkn2a* loss. This data has been added to Figure 3 —figure supplement 1. As you can see, neurospheres from *Cdkn2a-/-* mice have increased NIC capacity to neurospheres from WT mice. As mentioned above, our results are corroborated by previous studies from other studies (Krishnamurthy et al., 2004; Molofsky et al., 2006; Zindy et al., 1997).

4. In the Tg-SwDI/*Usp16*^+/-^ is Bmi-1 expression changed (perhaps as a compensatory mechanism of *Usp16* loss). Could this explain the only partial rescue of NIC frequency (Figure 3F).

In microarray data we collected from the cortex of 2 year old mice, *Bmi1* expression was reduced in Tg-SwDI mice, which is rescued by USP16 haploinsufficiency, in both the SVZ and the cortex. This data is shown in Figure 3 —figure supplement 1.

5. In Figure 5E, is this quantification from the ELDA assay or a different assay? If different, ELDA should be used to remain consistent.

The nature of our ELDA assay is not conducive to effective dosing of 1-5 cells in a 96 well format every single day. Typically, ELDA studies with drug dosing such as that in Gao et al. are limited to dosing the cells before plating and not treating the cells during growth which limits the effectiveness of the study (Gao et al., 2019). We therefore opted to perform a colony assay instead which allowed us to effectively observe differences when testing different concentrations on wild type and mutant APP neurospheres and also allowed us to accurately dose the neurospheres. In Lee et al. for example, they acknowledge these challenges and create a similar clonogenic assay to the ones we use here (Lee et al., 2014).

6. While the authors show quantitative data of Thioflavin staining in Figure 6B, representative images should be included in figure or supplementary. Similarly, representative images of GFAP staining for Figure 6A should be included.

These have been added as part of Figure 6 —figure supplement 1.

7. The authors should be more overall more critical with their own data, and explain what aspects of their model system are likely useful and can be translated to a human situation, and which ones are not. I would be very hesitant to commit to what extent the mouse SVZ relates to the human situation. A phenotype detected in a 3-4 months-old transgenic mice might be translatable to early human development (fetal/early life), to adult neurogenesis in early AD pathology (late in life), or not directly translatable. Mouse and human NPCs are extremely different cells (see work from Conti L. et al., Sun et al., Elkabetz et al., etc…), and this is also eident from the data here (e.g. compare scales in Figure 1B and 1E). I suggest the authors to please avoid rushed over-interpretation.

We recognize that all of the findings in our work may not be translatable to the human situation and have tried to adapt the text to emphasize what may or may not be translatable, however, these animal studies would need to be followed up with human studies to affirm or deny any translatability. We have also addressed this point with more detail in our revised discussion.

One important feature of our work that we suggest is translatable to humans lays in that our model system is useful in detecting an earlier phenotype present either before or at the same time as the development of the amyloid β plaques, namely the NPC self-renewal defect. While there are inherent differences between the NPCs of mice and humans, the aberrant effects on the NPCs secondary to *APP* mutations may similarly impact neurodegeneration and cognitive defects the mice develop later on that are not attenuated with amyloid-ß reduction. The timing of therapeutic treatment in Alzheimer’s disease seems to be crucial as studies have shown that treating the disease too late has little efficacy (Sperling et al., 2011; Yiannopoulou, Anastasiou, Zachariou, and Pelidou, 2019). We therefore hypothesize that when plaques are first visible in the patient, therapeutic reduction of *Usp16* or BMP signaling may reduce the defect contributing to symptomatic AD later on in life, as seen in our mouse model. Of course, these studies would need to be performed on humans before making this claim.

8. It is not clear how the authors come to the conclusion that this "suggests that the self-renewal defect […] not specific to the Swedish, Dutch and Iowa mutations, but also more broadly seen with other APP mutations" (Figure 1E/p5). They did not test different expression/protein levels of wt, the individual mutations, or combinations, which would be necessary to make such a statement.

In the text, we have fixed this to simply state the two relevant models we have focused on: one with Swedish, Dutch, and Iowa mutations and the other with Swedish and Indiana mutations so as to not make any overstatements. Our statement was meant to highlight that with more than one type of *APP* mutation, we see a similar effect in NPCs. In this study, it would not be particularly useful to compare levels of expressed *APP* in WT and transgenic mice as a human transgene is required in this disease model; mutations in the endogenous mouse *App* gene do not produce clinical disease. However, we did test different expression levels of endogenous and mutant APP in both the human and murine cells, which are shown in response to reviewer point #9 below.

9. The attempt for assessing the effect of mutant APP on NPC phenotypes in the human system is a strength of this study. In Figure 1E/p5 the authors claim that "the effects observed in NPCs derived from a genetic mouse model can be robustly recapitulated in human NPCs expressing mutant APP". As to the APP setting, both models are very different, in that human cells possess two copies of endogenous APP. "Fetal human NPCs" are not very well defined, an no characterization of endogenous/transgenic APP expression/protein levels are provided here. The choice of this models seems suboptimal. Also, to draw meaningful conclusions, the effect should be tested on different genetic backgrounds and well-characterized NPCs (e.g. from APP-mutant iPSCs). The conclusion that the "effects observed in NPCs derived from a genetic mouse model can be robustly recapitulated in human NPCs" is thus not appropriate and probably vastly oversimplified.

We agree with the reviewer that our conclusion is oversimplified so we have appropriately corrected our choice of words to describe these results in our text. In selecting a human model, we wanted to choose primary cells that were easily available to us and would limit other potential sources of variation (such as different genetic backgrounds). As the reviewer stated, one strength of this study is our attempt to show effect of mutant APP on NPC phenotype. The transduction of the human fetal NPCs with either ZsGreen or mutant APP narrows down the phenotypes we see to solely the effects of the mutant APP. To further characterize our system, we show in Figure 1 —figure supplement 1 that three different transductions of human fetal NPCs have resulted in at least a 2 fold increase in *APP* expression in neurospheres compared to ZsGreen controls.

To provide a characterization of mutant APP in our mouse model, we can turn to our sc-RNA-seq data where we conducted a differential expression gene test and found an ~1.5 fold change between our Tg-SwDI and WT cells in both transgenic (labeled “APPSDI”) and endogenous *App* expression (also shown in Figure 4 —figure supplement 2). Our conclusion can thus be amended to say that the decrease in neurosphere formation in NPCs derived from our genetic mouse model has also been observed in our human fetal model with roughly the same fold increase in mutant APP expression.

10. The authors show that the neurosphere phenotype of NPCs of 3-month-old mice that was rescued by Cdkn2a-KO. This is followed by the statement (line158) that APP mutations accelerate aging through Cdkn2a. In the context pf the presented data, this is a prominent overstatement.

We have updated the manuscript to more accurately depict our findings, and the text now reads, “These results demonstrate that impairment of NPC regeneration, as measured by NIC frequencies, is a function of aging that is accelerated by *APP* mutations and is mitigated through loss of *Cdkn2a…”*

References:

Barbar, L., Jain, T., Zimmer, M., Kruglikov, I., Sadick, J. S., Wang, M.,... Fossati, V. (2020). CD49f Is a Novel Marker of Functional and Reactive Human iPSC-Derived Astrocytes. Neuron, 107(3), 436-453 e412. DOI: 10.1016/j.neuron.2020.05.014.

Bateman, R. J., Xiong, C., Benzinger, T. L., Fagan, A. M., Goate, A., Fox, N. C.,... Dominantly Inherited Alzheimer, N. (2012). Clinical and biomarker changes in dominantly inherited Alzheimer's disease. N Engl J Med, 367(9), 795-804. DOI: 10.1056/NEJMoa1202753.

Bruggeman, S. W., Valk-Lingbeek, M. E., van der Stoop, P. P., Jacobs, J. J., Kieboom, K., Tanger, E.,... van Lohuizen, M. (2005). Ink4a and Arf differentially affect cell proliferation and neural stem cell self-renewal in Bmi1-deficient mice. Genes Dev, 19(12), 1438-1443. DOI: 10.1101/gad.1299305.

Gao, L., Huang, S., Zhang, H., Hua, W., Xin, S., Cheng, L.,... Pei, G. (2019). Suppression of glioblastoma by a drug cocktail reprogramming tumor cells into neuronal like cells. Sci Rep, 9(1), 3462. DOI: 10.1038/s41598-019-39852-5.

Jack, C. R., Jr., Knopman, D. S., Jagust, W. J., Shaw, L. M., Aisen, P. S., Weiner, M. W.,... Trojanowski, J. Q. (2010). Hypothetical model of dynamic biomarkers of the Alzheimer's pathological cascade. Lancet Neurol, 9(1), 119-128. DOI: 10.1016/S1474-4422(09)70299-6.

Krishnamurthy, J., Torrice, C., Ramsey, M. R., Kovalev, G. I., Al-Regaiey, K., Su, L., and Sharpless, N. E. (2004). Ink4a/Arf expression is a biomarker of aging. J Clin Invest, 114(9), 1299-1307. DOI: 10.1172/JCI22475.

Kumari, R., and Jat, P. (2021). Mechanisms of Cellular Senescence: Cell Cycle Arrest and Senescence Associated Secretory Phenotype. Front Cell Dev Biol, 9, 645593. DOI: 10.3389/fcell.2021.645593.

Lee, D. W., Choi, Y. S., Seo, Y. J., Lee, M. Y., Jeon, S. Y., Ku, B., and Nam, D. H. (2014). High-throughput, miniaturized clonogenic analysis of a limiting dilution assay on a micropillar/microwell chip with brain tumor cells. Small, 10(24), 5098-5105. DOI: 10.1002/smll.201401074.

Molofsky, A. V., He, S., Bydon, M., Morrison, S. J., and Pardal, R. (2005). Bmi-1 promotes neural stem cell self-renewal and neural development but not mouse growth and survival by repressing the p16Ink4a and p19Arf senescence pathways. Genes Dev, 19(12), 1432-1437. DOI: 10.1101/gad.1299505.

Molofsky, A. V., Pardal, R., Iwashita, T., Park, I. K., Clarke, M. F., and Morrison, S. J. (2003). Bmi-1 dependence distinguishes neural stem cell self-renewal from progenitor proliferation. Nature, 425(6961), 962-967. DOI: 10.1038/nature02060.

Molofsky, A. V., Slutsky, S. G., Joseph, N. M., He, S., Pardal, R., Krishnamurthy, J.,... Morrison, S. J. (2006). Increasing p16INK4a expression decreases forebrain progenitors and neurogenesis during ageing. Nature, 443(7110), 448-452. DOI: 10.1038/nature05091.

Morris, J. C., Roe, C. M., Grant, E. A., Head, D., Storandt, M., Goate, A. M.,... Mintun, M. A. (2009). Pittsburgh compound B imaging and prediction of progression from cognitive normality to symptomatic Alzheimer disease. Arch Neurol, 66(12), 1469-1475. DOI: 10.1001/archneurol.2009.269.

Salloway, S., Sperling, R., Fox, N. C., Blennow, K., Klunk, W., Raskind, M.,... Clinical Trial, I. (2014). Two phase 3 trials of bapineuzumab in mild-to-moderate Alzheimer's disease. N Engl J Med, 370(4), 322-333. DOI: 10.1056/NEJMoa1304839.

Sperling, R. A., Aisen, P. S., Beckett, L. A., Bennett, D. A., Craft, S., Fagan, A. M.,... Phelps, C. H. (2011). Toward defining the preclinical stages of Alzheimer's disease: recommendations from the National Institute on Aging-Alzheimer's Association workgroups on diagnostic guidelines for Alzheimer's disease. Alzheimers Dement, 7(3), 280-292. DOI: 10.1016/j.jalz.2011.03.003.

Villemagne, V. L., Pike, K. E., Chetelat, G., Ellis, K. A., Mulligan, R. S., Bourgeat, P.,... Rowe, C. C. (2011). Longitudinal assessment of Abeta and cognition in aging and Alzheimer disease. Ann Neurol, 69(1), 181-192. DOI: 10.1002/ana.22248.

Yiannopoulou, K. G., Anastasiou, A. I., Zachariou, V., and Pelidou, S. H. (2019). Reasons for Failed Trials of Disease-Modifying Treatments for Alzheimer Disease and Their Contribution in Recent Research. Biomedicines, 7(4). DOI: 10.3390/biomedicines7040097.

Zamanian, J. L., Xu, L., Foo, L. C., Nouri, N., Zhou, L., Giffard, R. G., and Barres, B. A. (2012). Genomic analysis of reactive astrogliosis. J Neurosci, 32(18), 6391-6410. DOI: 10.1523/JNEUROSCI.6221-11.2012.

Zindy, F., Quelle, D. E., Roussel, M. F., and Sherr, C. J. (1997). Expression of the p16INK4a tumor suppressor versus other INK4 family members during mouse development and aging. Oncogene, 15(2), 203-211. DOI: 10.1038/sj.onc.1201178.